# Hydrogen-based metabolism as an ancestral trait in lineages sibling to the Cyanobacteria

Paula B. Matheus Carnevali [1], Frederik Schulz [2], Cindy J. Castelle[1], Rose S. Kantor[1,13], Patrick M. Shih [3,4,14], Itai Sharon[1,15,16], Joanne M. Santini [5], Matthew R. Olm[6], Yuki Amano [7,8], Brian C. Thomas[1], Karthik Anantharaman [1,17], David Burstein [1,18], Eric D. Becraft[19,9], Ramunas Stepanauskas [9], Tanja Woyke [2] & Jillian F. Banfield [1,6,10,11,12]

The evolution of aerobic respiration was likely linked to the origins of oxygenic Cyanobacteria. Close phylogenetic neighbors to Cyanobacteria, such as Margulisbacteria (RBX-1 and ZB3), Saganbacteria (WOR-1), Melainabacteria and Sericytochromatia, may constrain the metabolic platform in which aerobic respiration arose. Here, we analyze genomic sequences and predict that sediment-associated Margulisbacteria have a fermentation-based metabolism featuring a variety of hydrogenases, a streamlined nitrogenase, and electron bifurcating complexes involved in cycling of reducing equivalents. The genomes of ocean-associated Margulisbacteria encode an electron transport chain that may support aerobic growth. Some Saganbacteria genomes encode various hydrogenases, and others may be able to use $O_2$ under certain conditions via a putative novel type of heme copper $O_2$ reductase. Similarly, Melainabacteria have diverse energy metabolisms and are capable of fermentation and aerobic or anaerobic respiration. The ancestor of all these groups may have been an anaerobe in which fermentation and $H_2$ metabolism were central metabolic features. The ability to use $O_2$ as a terminal electron acceptor must have been subsequently acquired by these lineages.

[1] Department of Earth and Planetary Science, University of California, Berkeley, Berkeley 94720 CA, USA. [2] DOE Joint Genome Institute, Walnut Creek 94598 CA, USA. [3] Feedstocks Division, Joint BioEnergy Institute, Emeryville 94608 CA, USA. [4] Environmental Genomics and Systems Biology Division, Lawrence Berkeley National Laboratory, Berkeley 94720 CA, USA. [5] Institute of Structural & Molecular Biology, Division of Biosciences, University College London, London WC1E 6BT, UK. [6] Department of Plant and Microbial Biology, University of California, Berkeley, Berkeley 94720 CA, USA. [7] Nuclear Fuel Cycle Engineering Laboratories, Japan Atomic Energy Agency, Tokai 319-1111 Ibaraki, Japan. [8] Horonobe Underground Research Center, Japan Atomic Energy Agency, Horonobe 098-3224 Hokkaido, Japan. [9] Bigelow Laboratory for Ocean Sciences, East Boothbay 04544 ME, USA. [10] Chan Zuckerberg Biohub, San Francisco 94158 CA, USA. [11] Earth Sciences Division, Lawrence Berkeley National Laboratory, Berkeley 94705 CA, USA. [12] Innovative Genomics Institute, Berkley 94704 CA, USA. [13]Present address: Department of Civil and Environmental Engineering, University of California, Berkeley, Berkeley 94720 CA, USA. [14]Present address: Department of Plant Biology, University of California, Davis, Davis 95616 CA, USA. [15]Present address: Migal Galilee Research Institute, Kiryat Shmona 11016, Israel. [16]Present address: Tel Hai College, Upper Galilee 12210, Israel. [17]Present address: Department of Bacteriology, University of Wisconsin-Madison, Madison 53706 WI, USA. [18]Present address: School of Molecular and Cell Biology and Biotechnology, George S. Wise Faculty of Life Sciences, Tel Aviv University, Tel Aviv 69978, Israel. [19]Present address: Department of Biological Sciences, North Alabama University, Florence 35632 AL, USA. These authors contributed equally: Paula B. Matheus Carnevali, Frederik Schulz  Correspondence and requests for materials should be addressed to J.F.B. (email: jbanfield@berkeley.edu)

Oxygenic photosynthesis evolved in Cyanobacteria and ultimately led to the Great Oxidation Event (GOE) ~2.3 billion years ago[1]. This was a major step in the co-evolution of life and the planet[2]. The acquisition of the combination of photosystem I and photosystem II by Cyanobacteria allowed exploitation of water ($H_2O$) as an electron donor and the production of oxygen ($O_2$). The transport of electrons through the chain of photosystem complexes enables export of protons ($H^+$) and generation of a proton motive force (PMF is electrochemical potential across a cell membrane that can be harnessed to form adenosine triphosphate— ATP). Formation of ATP is described as energy conservation because ATP is required for reactions such as carbon dioxide ($CO_2$) fixation, nitrogen ($N_2$) fixation and biosynthesis. Constitutive expression of fermentation pathways in Cyanobacteria[3] suggest that prior to the advent of oxygenic photosynthesis, the ancestral mechanism for ATP formation in these organisms was fermentation, which does not require a separate electron acceptor.

Following the GOE, organisms had access to higher energetic yield from aerobic respiratory processes that involve coupling oxidation of an electron donor such as organic carbon to reduction of $O_2$. Also following the GOE, the potential for anaerobic respiration would have greatly increased due to the availability of oxidized compounds, such as nitrate ($NO_3^-$) that can also serve as an electron acceptor ($NO_3^-$ forms in the environment mostly via $O_2$-dependent reactions). Aerobic respiration and $NO_3^-$ reduction required the evolution of redox complexes of the electron transport chain (ETC), and use of $O_2$ required the evolution of a terminal oxidase. The simplest ETC is composed of (i) an electron entry point, usually a dehydrogenase that oxidizes reduced electron carriers (e.g., reduced nicotinamide adenine nucleotide – NADH and succinate), (ii) membrane electron carriers (e.g., quinones), and (iii) an electron exit point such a heme-copper oxygen reductase or a cytochrome *bd* oxidase. However, if there are periplasmic electron donors, an intermediary quinol:electron acceptor oxidoreductase complex may also be involved[4]. Both the electron entry and exit protein complexes often act as $H^+$ pumps, contributing to the creation of a PMF.

The metabolic potential of lineages phylogenetically related to Cyanobacteria is of great interest from the perspective of constraining the biological context in which complex ETCs evolved. Recently, several publications have investigated the biology of Melainabacteria and Sericytochromatia, groups that branch adjacent to Cyanobacteria and are represented by non-photosynthetic organisms[5–7]. It has been suggested that the machinery for aerobic respiration was acquired independently by Sericytochromatia[7], and like Cyanobacteria, some members of the Sericytochromatia and Melainabacteria have ETCs. Analysis of genomes of additional major groups of bacteria sibling to the Cyanobacteria may help distinguish the possibilities that Melainabacteria and Sericytochromatia acquired anaerobic metabolisms after their divergence from Cyanobacteria from the alternative, in which Cyanobacteria gained aerobic metabolism after their divergence from Melainabacteria and Sericytochromatia.

Here we investigate genomes of Margulisbacteria (RBX-1) and Saganbacteria (WOR-1), lineages related to both Melainabacteria and Sericytochromatia, and identify common mechanisms for energy conservation that may have been present in their common ancestor with Cyanobacteria. Several genomes were previously reconstructed from estuarine sediments and groundwater[8–10]. For Margulisbacteria, we describe two groups that we refer to as Riflemargulisbacteria (given the derivation of draft genome sequences from the Rifle research site)[8], and four groups we refer to as Marinamargulisbacteria (given the derivation of the single-

cell draft genomes from ocean sites). Marinamargulisbacteria was first observed in a clone library of SSU rRNA genes from Zodletone spring (Oklahoma) and identified as candidate division ZB3[11]. We leverage these genomes, and new genomes for Margulisbacteria and members of the Melainabacteria to predict how organisms from these lineages conserve energy in the form of ATP, identify their potential strategies for reoxidation of reducing equivalents, and propose the interconnectedness between $H_2$ and $O_2$ metabolism. Membrane-bound complexes that may be involved in ion translocation for generation of an electrochemical membrane potential were of particular interest. We found that $H_2$ metabolism is a common feature of the modern groups that cannot respire aerobically, and suggest that hydrogenases may have been central to the lifestyles of the ancestors of Cyanobacteria, Melainabacteria, Sericytochromatia, Margulisbacteria, and Saganbacteria.

## Results

**Identification of representative genomes from each lineage**. In this study, we analyzed four publicly available and eight newly reconstructed genomes of Margulisbacteria and propose two distinct clades within this lineage: Riflemargulisbacteria (two groups) and Marinamargulisbacteria (four groups). These clades were defined using metagenome-assembled genomes (MAGs) of bacteria from the sediments of an aquifer adjacent to the Colorado River, Rifle, USA and single-cell amplified genomes (SAGs) from four ocean environments (Supplementary Data 1). We also analyzed 26 publicly available MAGs of Saganbacteria from groundwater with variable $O_2$ concentrations[8], of which four genomes are circularized. In addition, we analyzed eight publicly available MAGs of Melainabacteria[8], and five new Melainabacteria genomes that were reconstructed from metagenomic datasets for microbial communities sampled from human gut and groundwater (Supplementary Data 1). We identified representative genomes for groups of genomes that share 95.0–99.0% average nucleotide identity (ANI) (Supplementary Data 1) and focus our discussion on these genomes. Metabolic predictions for genomes that were estimated to be medium or high quality[12] (Supplementary Data 1) were established using gene annotations and confirmed using Hidden Markov Models (HMMs) built from the KEGG database (Supplementary Data 2). Key aspects of energy metabolism predicted for Riflemargulisbacteria, Marinamargulisbacteria, Saganbacteria, and Melainabacteria were compared. Phylogenetic analyses of genes encoding key metabolic functions also included publicly available genomes for Sericytochromatia[7], newly reported genomes for relatives of Marinamargulisbacteria[13], and other reference genomes.

**Relatives of Cyanobacteria, Melainabacteria, and Sericytochromatia**. In our phylogenetic analysis, Margulisbacteria and Saganbacteria group together in sibling position to a monophyletic clade consisting of Sericytochromatia, Melainabacteria and Cyanobacteria (Fig. 1a, b). In addition, Margulisbacteria, Saganbacteria and Sericytochromatia are positioned basally to Melainabacteria and Cyanobacteria. Importantly, the affiliation of these groups with the Cyanobacteria is consistent for phylogenetic trees based on 56 universal single copy proteins and 16 ribosomal proteins, which are a subset of the 56 universal single copy proteins (Fig. 1a, Supplementary Fig. 1, 2). However, the precise position of these three lineages in the Terrabacteria differs depending on the set of markers and phylogenetic models used for tree construction (Fig. 1a, Supplementary Fig. 2) and the deep branching order often cannot be well resolved[13,14]. In addition, the position of Margulisbacteria, Saganbacteria, and Cyanobacteria relative to the Thermoanaerobacterales (Fig. 1a)

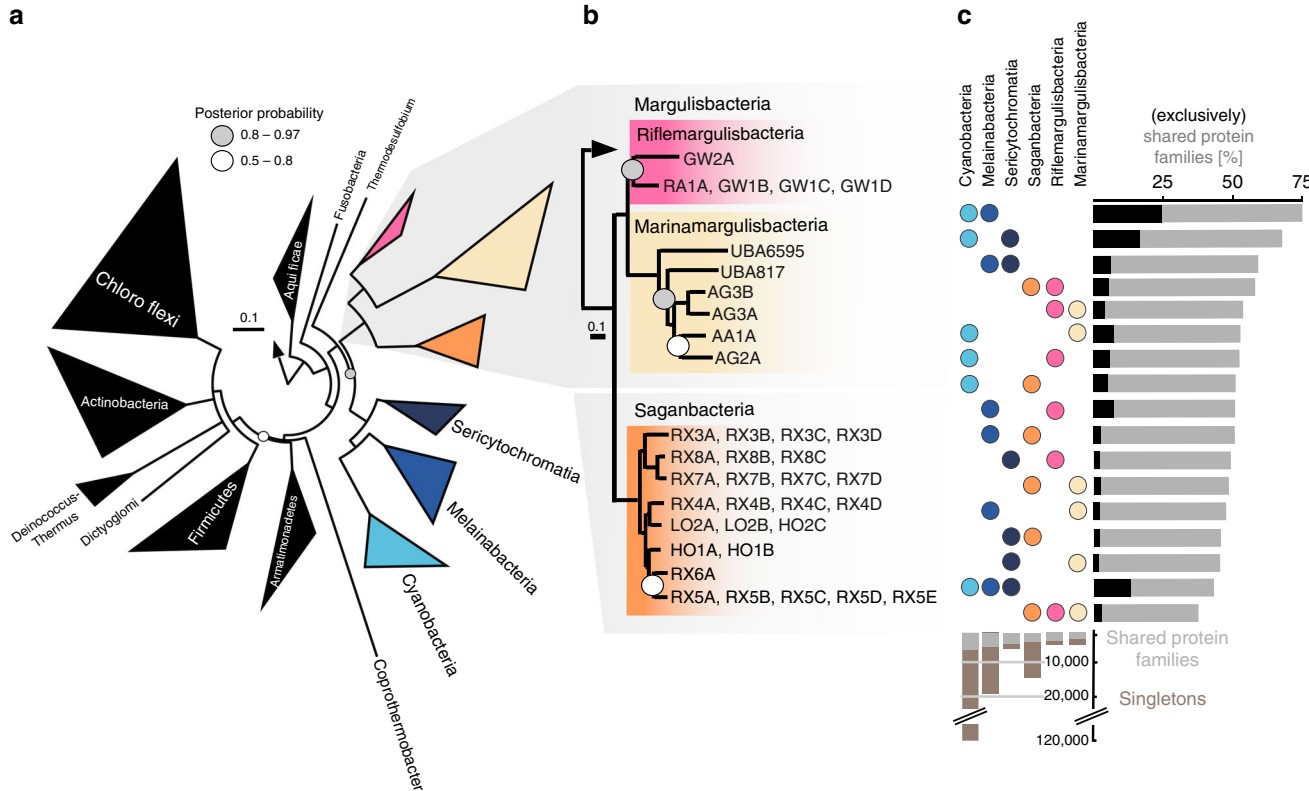

**Fig. 1** Phylogenetic tree based on 56 concatenated proteins and shared protein families. **a** Location of Margulisbacteria and Saganbacteria in a de-replicated species tree of the Terrabacteria (detailed tree shown as Supplementary Fig. 1). **b** Phylogenetic relationship of members of the Margulisbacteria and Saganbacteria. See Supplementary Data 1 for genome identification details. Support values are not shown at nodes with maximal support (posterior probabilities > 0.97). **c** Shared protein families among lineages: rows cover the total number of protein families shared among lineages and dots indicate which lineages are being compared. For example, the top bar indicates that in average 75% of protein families are shared between Cyanobacteria and Melainabacteria, out of which 25% are exclusively shared between them. The other 50% includes proteins that are shared with at least one other lineage. At the bottom, bars show for each lineage the total number of singletons (unique to respective lineage) in brown, and the number of protein families that is shared with one or more lineages in gray. One hundred percent shared represents the union of all the non-singleton families of the two lineages under comparison. For details on taxon sampling, tree inference and shared protein families, see Methods section

will require further investigation once more genomes for these groups become available. Similar to the monophyly of Sericytochromatia, Cyanobacteria, and Melainabacteria, the monophyly of Saganbacteria, Riflemargulisbacteria and Marinamargulisbacteria is well supported (posterior probability >0.97). The analysis of shared protein families among lineages, suggested that Riflemargulisbacteria, Marinamargulisbacteria, and Saganbacteria share less protein families than Cyanobacteria, Melainabacteria and Sericytochromatia do (Fig. 1c). Despite the closer phylogenetic relationship between Riflemargulisbacteria and Marinamargulisbacteria, Riflemargulisbacteria share a greater proportion of protein families with Saganbacteria than with Marinamargulisbacteria, whereas Marinamargulisbacteria share most protein families with the more distantly related Cyanobacteria. Taken together, these findings may indicate niche adaptation and a divergent lifestyle of Riflemargulisbacteria and Marinamargulisbacteria.

**$H_2$ metabolism may be vital for generating a proton-motive force.** Metabolic analyses based on a representative genome of the sediment-associated Riflemargulisbacteria, Margulisbacteria RA1A (96% completeness, representative of four genomes in the same ANI cluster, see Supplementary Data 1), suggest that members of Riflemargulisbacteria are heterotrophic organisms with a fermentation-based metabolism (Fig. 2). They use

hydrogenases and other electron bifurcating complexes to balance reducing equivalent pools (NADH and ferredoxin), and rely on membrane-bound protein complexes to generate a proton/sodium ($H^+/Na^+$) potential and to make ATP (Fig. 3a). We predict that Margulisbacteria RA1A is an obligate anaerobe, and multiple enzymes that participate in central metabolism use ferredoxin as the preferred electron donor/acceptor (Fig. 2). Furthermore, Margulisbacteria RA1A lacks most components of the tricarboxylic acid (TCA) cycle and an ETC (Supplementary Data 2), including terminal reductases for aerobic or anaerobic respiration. Carbon dioxide ($CO_2$) fixation pathways were not identified in genomes of this lineage (Supplementary Data 2). For further details about the central metabolism, lifestyle, and other features of Riflemargulisbacteria see Supplementary Notes 1, 2, 7 and 8 and Supplementary Data 3 and 4.

Hydrogen metabolism appears essential for the sediment-associated Riflemargulisbacteria. In the absence of a well-defined respiratory electron transport chain, reduced ferredoxin and NADH can be re-oxidized by reducing $H^+$ to hydrogen ($H_2$). Hydrogenases, the key enzymes in hydrogen metabolism, can be reversible and either use $H_2$ as a source of reducing power or $H^+$ as oxidants to dispose of excess reducing equivalents. The assignments of hydrogenase types in this organism were based on phylogenetic analysis of the hydrogenase catalytic subunits (Fig. 4 and Supplementary Fig. 4), analysis of the subunit composition and predicted protein domains. Notably, only NiFe hydrogenases

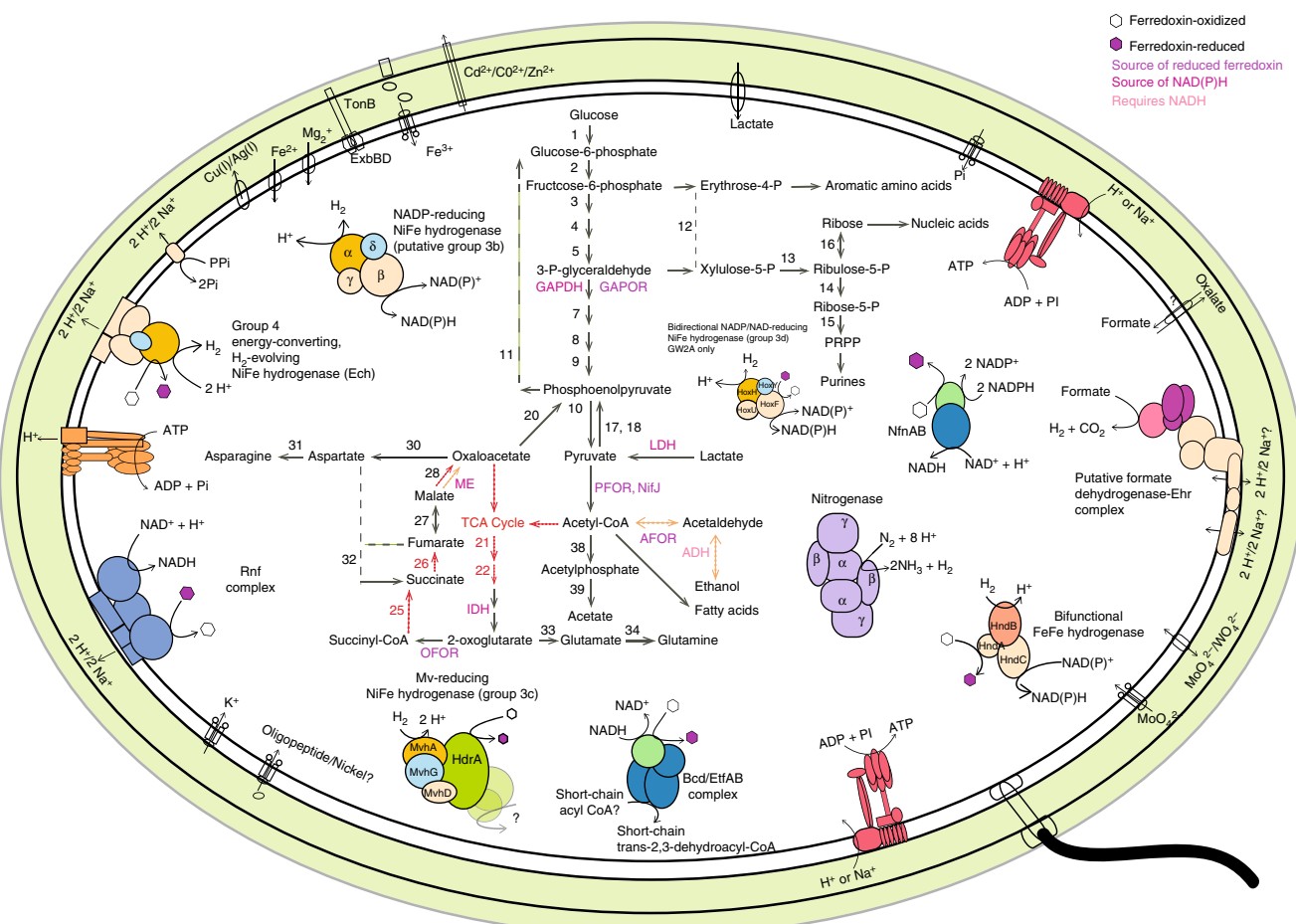

**Fig. 2** Margulisbacteria RA1A cell cartoon. Key enzymes predicted to be involved in core metabolic pathways include: (1) glucokinase, (2) glucose-6-phosphate isomerase, (3) 6-phosphofructokinase 1, (4) fructose-bisphosphate aldolase class I and II, (5) triosephosphate isomerase, *GAPDH* glyceraldehyde 3-phosphate dehydrogenase, *GAPOR* glyceraldehyde-3-phosphate dehydrogenase (ferredoxin), (7) phosphoglycerate kinase, (8) phosphoglycerate mutase 2,3-bisphosphoglycerate-dependent and 2,3-bisphosphoglycerate-independent, (9) enolase, (10) pyruvate kinase, (11) fructose-1,6-bisphosphatase III, (12) transketolase, (13) ribulose-phosphate 3-epimerase, (14) ribose 5-phosphate isomerase B, (15) ribose-phosphate pyrophosphokinase, (16) ribokinase, (17) pyruvate, orthophosphate dikinase, (18) pyruvate water dikinase, LDH: lactate dehydrogenase, (20) phosphoenolpyruvate carboxykinase (ATP), (21) citrate synthase, (22) aconitate hydratase, *IDH* isocitrate dehydrogenase, *OFOR* 2-oxoglutarate ferredoxin oxidoreductase, (25) succinyl-CoA synthetase, (26) succinate dehydrogenase, (27) fumarate hydratase, (28) malate dehydrogenase, *ME* malic enzyme (oxaloacetate-decarboxylating), (30) aspartate transaminase, (31) asparagine synthetase, (32) ʟ-aspartate oxidase, (33) glutamate synthase, (34) glutamine synthetase, *PFOR* pyruvate:ferredoxin oxidoreductase also NifJ: pyruvate-ferredoxin/flavodoxin oxidoreductase, *AFOR* aldehyde:ferredoxin oxidoreductase, *ADH* alcohol dehydrogenase, (38) phosphate acetyltransferase, (39) acetate kinase

from groups 3 and 4 are encoded in the genome, and these seem to have been horizontally transferred based on phylogenetic analyses. Group 4 NiFe hydrogenases are thought to be among the most primitive respiratory complexes[15].

We identified three types of cytoplasmic NiFe hydrogenases (groups 3b, 3c, and 3d; Fig. 4a; Supplementary Data 5-8) and one type of cytoplasmic FeFe hydrogenase (Supplementary Fig. 4 and Supplementary Data 9-12), in addition to one type of membrane-bound group 4 NiFe hydrogenase, and a hydrogenase-related complex (Fig. 4b; Supplementary Data 13-15 and Supplementary Data 8). For a detailed description of cytoplasmic hydrogenases see Supplementary Note 3. Given the importance of membrane-bound protein complexes in the generation of a membrane potential in prokaryotic cells, we focused our attention on the potentially membrane-bound complexes of Riflemargulisbacteria. Membrane-bound group 4 NiFe hydrogenases and related proteins share a common ancestor with NADH:ubiquinone oxidoreductase (Nuo), a protein complex that couples NADH oxidation with $H^+$ or ion translocation[16]. This complex is a key component of respiratory and photosynthetic electron transport chains in other organisms[17,18].

The Riflemargulisbacteria are predicted to have a group 4 membrane-bound NiFe hydrogenase (Fig. 4b), also called energy-converting hydrogenase (Ech). These enzymes usually couple the oxidation of reduced ferredoxin to $H_2$ evolution under anaerobic conditions[19], can be reversible[20], and are typically found in methanogens and other anaerobic organisms where they function as primary proton pumps[21]. Genes for two other complexes are found in close proximity to the Ech-type hydrogenase, a V-type ATPase and a *Rhodobacter* nitrogen fixation (Rnf) electron transport complex (Fig. 3a). These protein complexes may also contribute to the generation of an electrochemical potential in Margulisbacteria RA1A. This is remarkable because both the Ech hydrogenase and the Rnf complex are capable of generating a transmembrane potential, and only a small number of bacteria have been observed to have both[22]. The putative V-type ATPase operon in Margulisbacteria RA1A (*atpEXABDIK*) resembles those of Spirochetes and Chlamydiales[23] (Supplementary

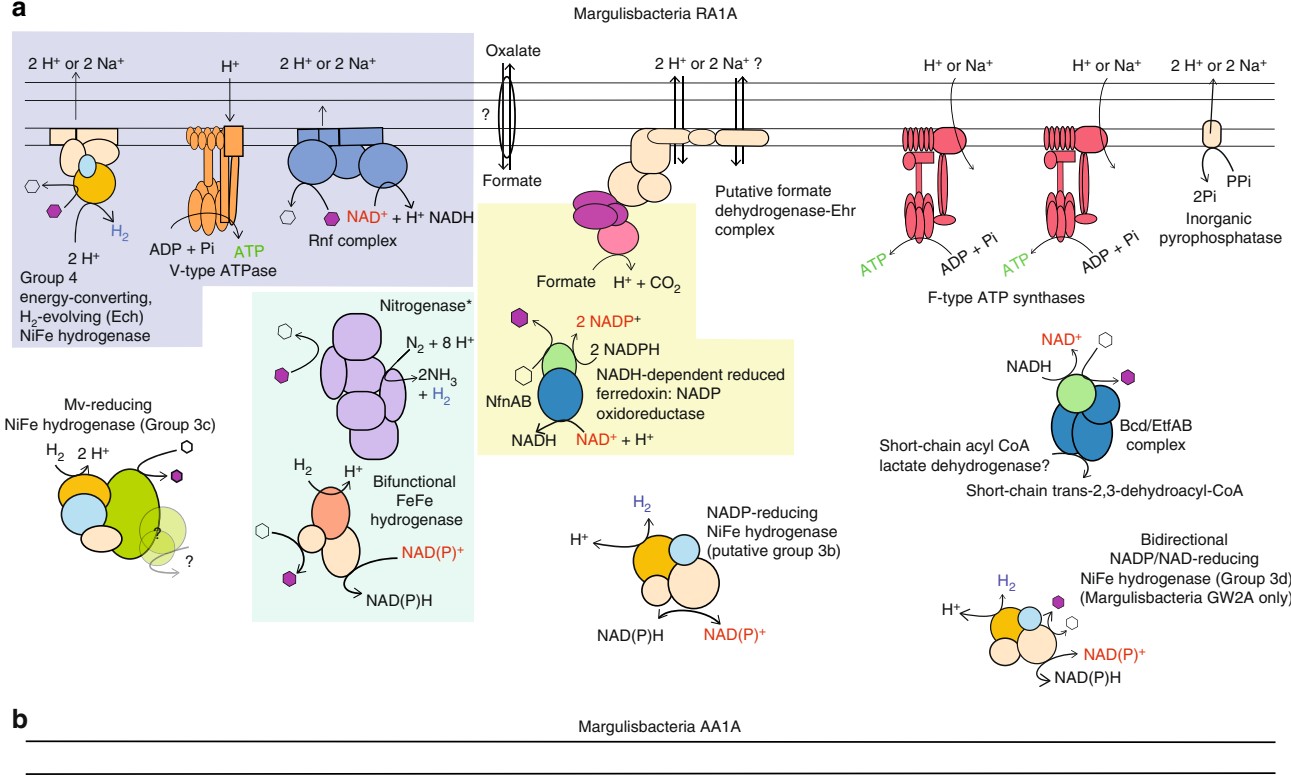

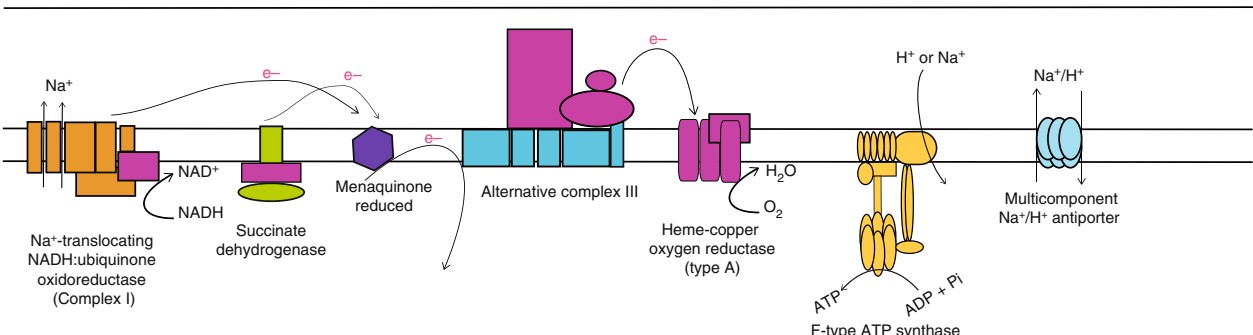

**Fig. 3** Schematic view of key protein complexes involved in energy metabolism. **a** Cytoplasmic and membrane bound hydrogenases, membrane-bound hydrogenase-related complex, ATPases, and electron bifurcating complexes in Margulisbacteria RA1A, representative of Riflemargulisbacteria. **b** Electron transport chain in Margulisbacteria AA1A, representative of Marinamargulisbacteria. Protein complexes encoded by genes in the same genomic region in the RA1A genome are highlighted as follows: the Ech NiFe hydrogenase, the V-type ATPase, and the Rnf complex in purple; the nitrogenase and the FeFe hydrogenase in blue; and the putative FDH and the Nfn complex in yellow

Data 16). This complex could hydrolyze or make ATP with $H^+$ or $Na^+$ translocation out of the cell. Likewise, the Rnf complex, which generally functions as a ferredoxin:$NAD^+$ oxidoreductase in *Rhodobacter capsulatum*, can couple an exergonic redox reaction to $Na^+$ translocation across the membrane[24]. The Rnf operon in Margulisbacteria RA1A is composed of six genes (*rnfCDGEAB*), similar in organization to the operon in *Acetobacterium woodii* and evolutionarily related to the Na$^+$-translocating NADH:quinone oxidoreductase (Nqr)[24,25].

Riflemargulisbacteria genomes also encode energy-converting hydrogenase-related complexes (Ehr), which are related to group 4 NiFe hydrogenases, but cannot be classified as such because they lack the conserved amino acids that coordinate Ni and Fe[16,26]. Ehr complexes are suspected to couple catalytic oxidation-reduction reactions in the cytoplasm to charge translocation across the membrane[16], due to the presence of multiple resistance and pH adaptation (Mrp) antiporter-like subunits[16]. For instance, in *Pyrococcus furiosus* a membrane-bound oxidoreductase (Mbx; a type of hydrogenase-related complex) functions as a ferredoxin: nicotinamide adenine nucleotide phosphate ($NADP^+$) oxidoreductase complex that potentially pumps $H^+$ across the membrane[27]. Given the Ehr subunit composition in Riflemargulisbacteria (Supplementary Fig. 3a and Supplementary Note 4) and the predicted need for an electron donor component, we looked for potential partners. In Riflemargulisbacteria, the candidate cytoplasmic oxidoreductase partner may be a molybdenum- or tungstate-dependent formate dehydrogenase (*fdhA*; Fig. 5; Supplementary Data 17), which likely oxidizes formate to $CO_2$ and NADPH, possibly reversibly. That $NAD(P)^+$ may be the electron acceptor for this reaction is inferred from the presence of two genes (*nuoE*- and *nuoF-like*) next to *fdhA*, encoding FeS cluster-binding domains and an electron carrier flavin mononucleotide (FMN)-binding domain (Supplementary Fig. 3b and Supplementary Note 4). Therefore, we hypothesize the presence of a formate dehydrogenase (FDH)-Ehr complex. Notably, genes encoding a full pathway for

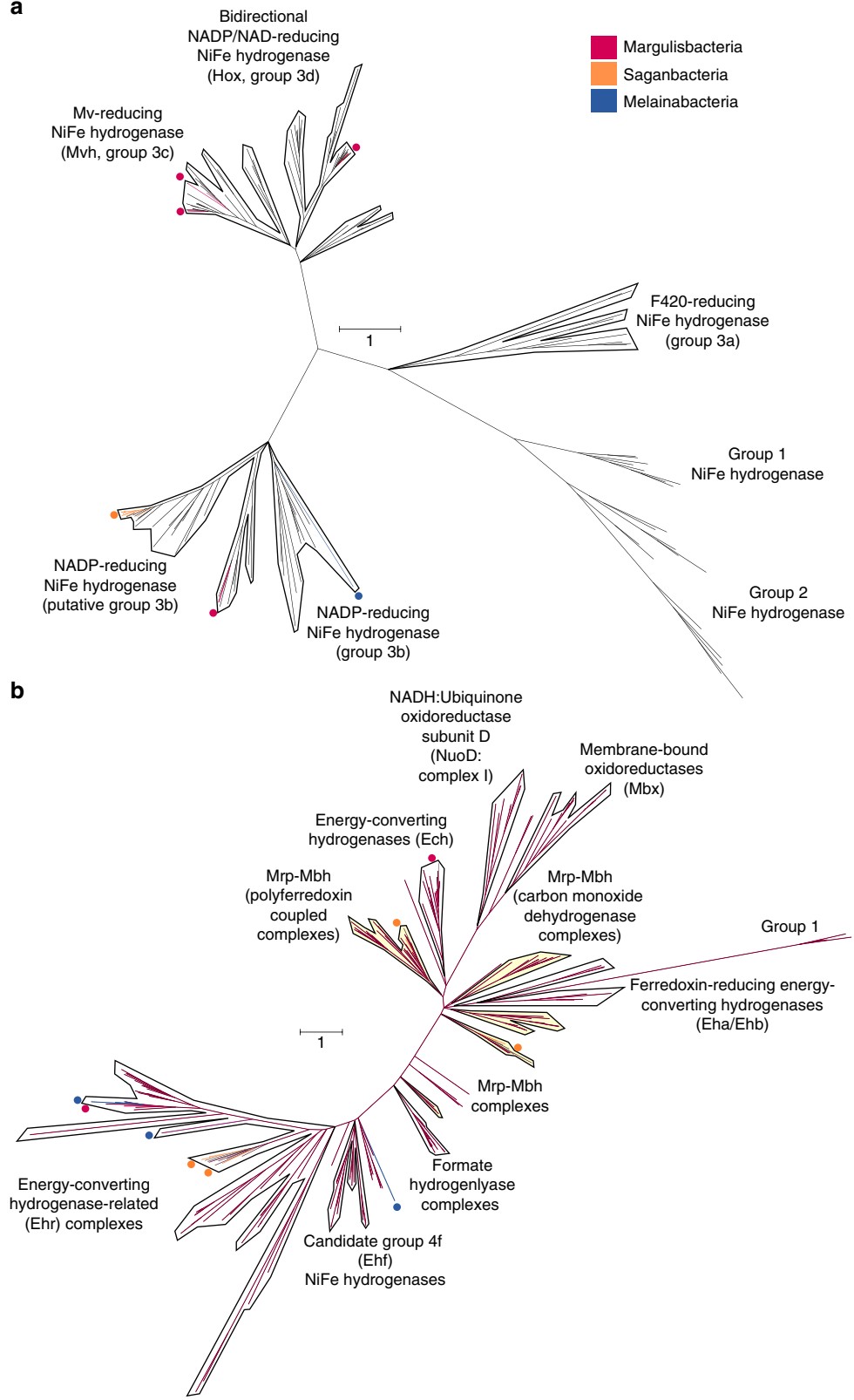

**Fig. 4** Phylogenetic tree of the catalytic subunit in NiFe hydrogenases. **a** Bayesian phylogeny indicating the phylogenetic relationships of the Riflemargulisbacteria, Saganbacteria, and Melainabacteria groups 1, 2 and 3 NiFe hydrogenase catalytic subunits and **b** Bayesian phylogeny indicating the phylogenetic relationships of the Riflemargulisbacteria, Saganbacteria, and Melainabacteria group 4 NiFe hydrogenases and related complexes. Scale bar indicates substitutions per site. Branches with a posterior support of below 0.5 were collapsed. The tree and underlying alignments are available with full bootstrap values as pdf and in Newick format in Supplementary Data 5-8 and 13-15

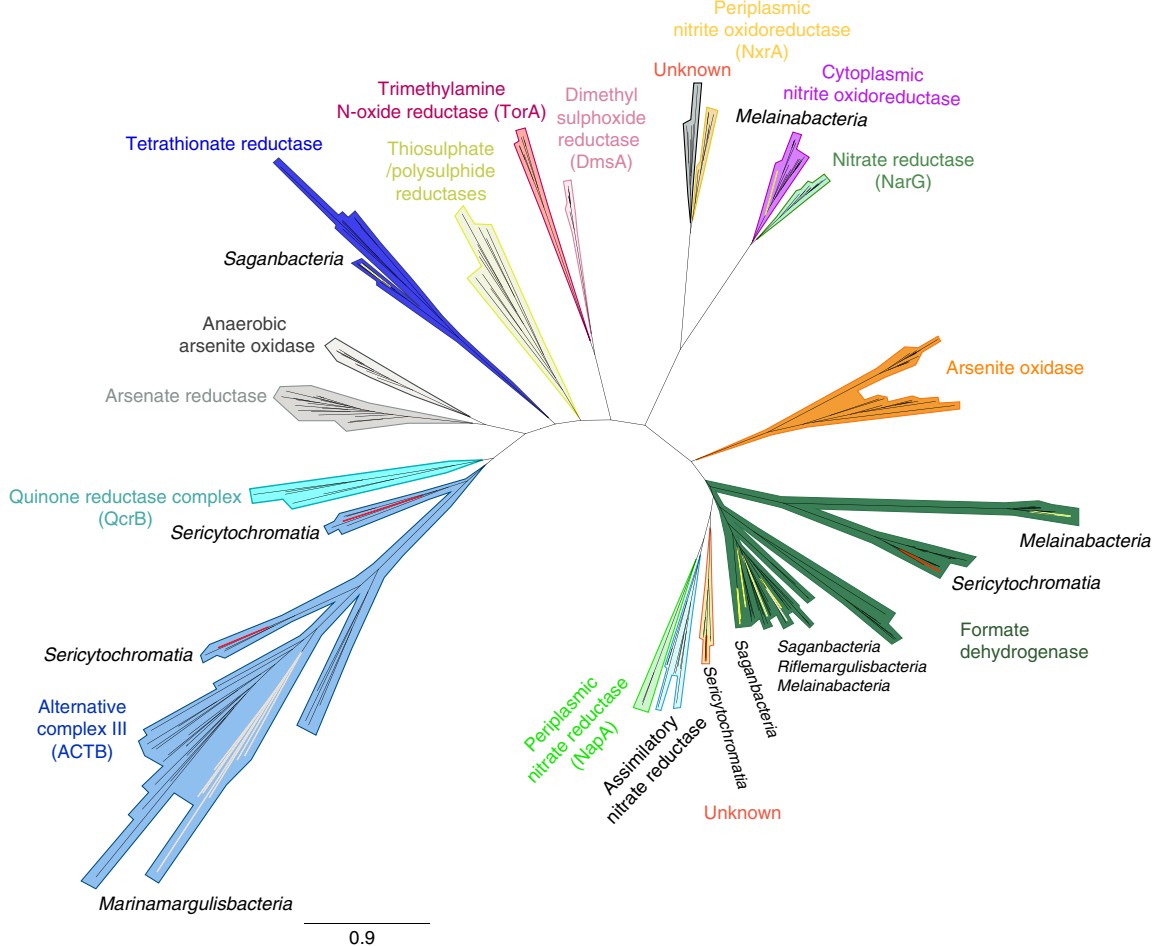

**Fig. 5** Phylogenetic analysis of the dimethyl sulfoxide (DMSO) reductase superfamily. Catalytic subunits of a formate dehydrogenase (FdhA) were found in Riflemargulisbacteria, Saganbacteria, and Melainabacteria (also present in some Cyanobacteria). An Alternative Complex III (ACIII; ActB subunit) was found in Marinamargulisbacteria, as well as in Sericytochromatia LSPB_72 and CBMW_12[7]. Subunit A (TtrA) of a tetrathionate reductase was identified in Saganbacteria. A cytoplasmic nitrate/nitrite oxidoreductase (NXR) was identified in Melainabacteria in this study. The tree is available with full bootstrap values in Newick format in Supplementary Data 17

synthesis of ubiquinone and menaquinone, which could serve as the electron acceptor for the overall reaction catalyzed by such complex were not identified in these genomes.

Lastly, Margulisbacteria RA1A encodes a nitrogenase, an enzyme known to produce $H_2$ during nitrogen fixation[28]. A phylogenetic analysis of key nitrogenase subunits NifHDK (Fig. 6; Supplementary Data 18-21) indicates that this enzyme is different from the nitrogenase in many Cyanobacteria. The nitrogenase operon (*nifHI$_1$I$_2$DKENBV*) in Riflemargulisbacteria resembles that of methanogens. Besides genes encoding the Fe protein (NifH) and the MoFe protein (NifDK), this operon includes genes involved in post-translational regulation (*nifI$_1$I$_2$*) and the synthesis of the enzyme (*nifENBV*). Additionally, *nifA* is found downstream from the operon, which is involved in transcriptional regulation. Therefore, this operon is of slightly higher complexity than the operon in methanogens (following[29]). Phylogenetically, NifHDK in Riflemargulisbacteria are related to Spirochetes, and other anaerobic taxa at the base of the tree (Fig. 6). Furthermore, we identified three genes that encode an FeFe hydrogenase (Supplementary Fig. 4) downstream of the *nif* operon. Given the proximity of the FeFe hydrogenase genes on the Margulisbacteria RA1A genome to the Nif operon, this enzyme may be responsible for dissipating excess $H_2$ generated by the nitrogenase complex involved in $N_2$ fixation and its expression could be regulated by nitrogen availability[30]. Based on the predicted subunits domain

composition, this enzyme could be classified as a group 2 (G2) bifurcating hydrogenase with a 3c modular structure[31]. In the forward direction, FeFe hydrogenases can bifurcate electrons from $H_2$ to ferredoxin and nicotinamide adenine dinucleotide phosphate –NAD(P)$^+$[19]. Interestingly, FeFe hydrogenases have not been found in Cyanobacteria (or Saganbacteria).

Like Margulisbacteria, Saganbacteria have anaerobic and aerobic representatives. Similar to sediment-associated Margulisbacteria, Saganbacteria include heterotrophic anaerobes that may generate a $H^+/Na^+$ potential via membrane-bound group 4 NiFe hydrogenases and hydrogenase-related complexes (Fig. 4b), in combination with an Rnf complex.

The studied genomes of Saganbacteria encode distinct types of hydrogenases. In the category of cytoplasmic complexes, we only identified one type, the group 3b NiFe hydrogenases (sometimes referred to as sulfhydrogenases), and these occurred in multiple representative genomes (Fig. 4a, Supplementary Data 22). Many Saganbacteria representative genomes also encode three distinct types of membrane-bound hydrogenases (Fig. 4b). Genomic regions including *fdhA* and *nuoE*-and *nuoF*-like genes also occur in many Saganbacteria with Ehr complexes (Supplementary Figs. 3b and c), an observation that strengthens the deduction that an FDH-Ehr complex may occur in these clades.

Another type of group 4 NiFe hydrogenase found in Saganbacteria RX5A is enigmatic. The most closely related

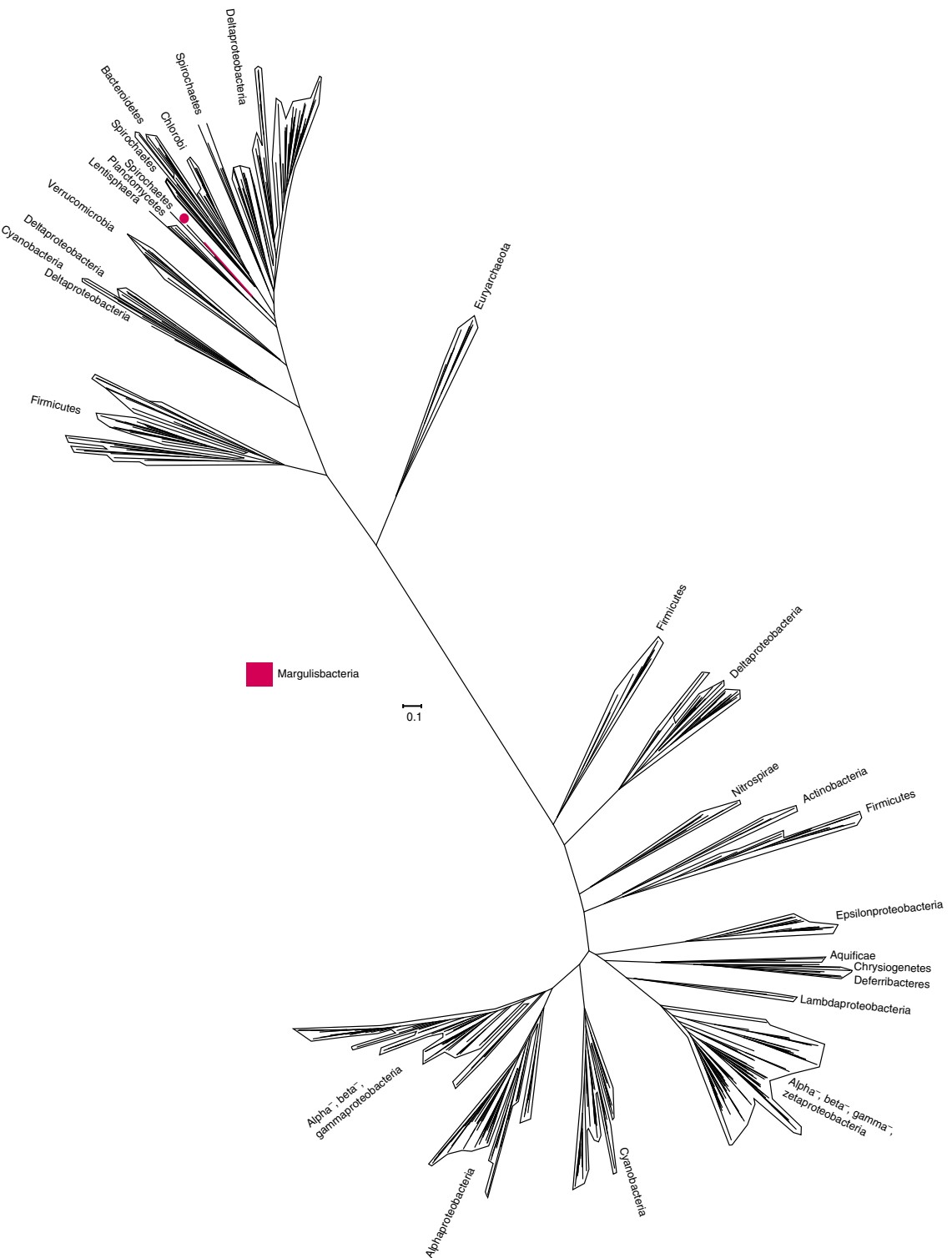

**Fig. 6** Phylogenetic tree of concatenated nitrogenase subunits NifHDK. Maximum likelihood tree indicating the branching of the Riflemargulisbacteria NifHDK together with anaerobic taxa. Scale bar indicates substitutions per site. Branches with a posterior support of below 0.5 were collapsed. The tree and underlying alignments are available with full bootstrap values as pdf and in Newick format in Supplementary Data 18-21

sequences are hydrogenases found in candidate phyla such as Zixibacteria and Omnitrophica (Fig. 4b, Supplementary Data 13). The genomic region encodes hydrogenase subunits, Mrp antiporter-like subunits, and a molybdopterin-containing oxidoreductase that could not be classified by phylogeny. Specifically, it could not be identified as a formate dehydrogenase, although the HMMs suggested this to be the case. Also present in the genomic

region are a putative CO dehydrogenase subunit F gene (*cooF*), and putative anaerobic sulfite reductase subunits A (*asrA*) and B (*asrB*) genes (Supplementary Fig. 3e and Supplementary Note 5). Together, these subunits might compose an oxidoreductase module. We hypothesize that the function of the proteins encoded in this region is similar to that of Mbh hydrogenases described in other organisms[32].

Two other putative group 4 NiFe hydrogenases were found in Saganbacteria RX6A (Supplementary Figs. 3d and f). Each shares some components with the Mbh hydrogenase described above. One of them has Mrp antiporter-like subunits, but lacks an oxidoreductase module, and it is phylogenetically most closely related to hydrogenases in *Pelobacter propionicus* and Clostridia (Fig. 4b, Supplementary Data 13 and Fig. 3f). Potentially this could be another Mrp-Mbh complex, and the lack of an oxidoreductase module combined with the presence of Fe–S cluster-binding domains indicates that it may be involved in ferredoxin reoxidation, as occurs in *P. furiosus*[32]. The other putative group 4 NiFe hydrogenase harbors an oxidoreductase module, seems to lack Mrp antiporter-like subunits, and could not be placed phylogenetically due to a truncated sequence encoding a putative NiFe hydrogenase large (catalytic) subunit (Supplementary Fig. 3d). Therefore, we could not determine whether it is a Mbh-type hydrogenase or another type of hydrogenase-related complex (*e.g.* Mbx).

The majority of anaerobic Saganbacteria also possess genes encoding a Rnf complex and a V/A-type ATPase (Supplementary Data 2). The V/A-type ATPases have a subunit composition different from that in Riflemargulisbacteria, and their genes are in operons resembling those found in other bacteria and archaea (Supplementary Data 16).

Given the importance of hydrogenases and overall similarity in metabolic capacities of Riflemargulisbacteria and Saganbacteria, we investigated newly available genomes for the Melainabacteria for evidence of yet unrecognized $H_2$ metabolism in this group. We found that some representative Melainabacteria genomes have bifurcating FeFe hydrogenases (modular groups 3a and 3c[31]) that are monophyletic with the FeFe hydrogenase in Margulisbacteria, albeit in a separate phylogenetic cluster (Supplementary Fig. 4). For more detail about cytoplasmic hydrogenases in Melainabacteria, see Supplementary Note 6.

Ehr complexes are present in four representative Melainabacteria genomes (Fig. 4b). The sequences in three of the four representative genomes cluster with sequences from Riflemargulisbacteria, Lentisphaerae, Spirochetes and *Acinetobacter* sp., but Melainabacteria BJ4A branches on its own (Fig. 4b). In addition to Ehr, Melainabacteria RX6A and BJ4A have an *fdhA* gene, and RX6A has genes encoding NuoE- and NuoF-like subunits just downstream of the *fdhA* gene. Interestingly, FdhA clusters with proteins found in the Cyanobacteria *Nostoc piscinale* and *Scytonema hofmannii* (Fig. 5). Melainabacteria HO7A is the only one with a putative group 4 f hydrogenase (after[19]), which is closely related to *Gracilibacteria* sp. and other anaerobic organisms.

Some Margulisbacteria RA1A, Saganbacteria and Melainabacteria have complexes predicted to be involved in electron bifurcation. For hydrogenase-based electron bifurcation, reduced ferredoxin and NAD(P)H (redox carriers with very different electrode potentials) are oxidized, coupled to the reduction of $H^+$ to form $H_2$[33]. This reaction can also work in the reverse as the electron bifurcation complex can be bidirectional. Electron bifurcation complexes allow balance of reduced and oxidized cofactors in anaerobic environments, where electron acceptors are limited[15].

In close proximity to the region encoding FdhA in Margulisbacteria RA1A and most Saganbacteria genomes, we identified genes encoding a NADH-dependent reduced ferredoxin: NADP oxidoreductase (Nfn) complex (Fig. 3a). Nfn complexes use 2 NADPH molecules to catalyze the reversible reduction of oxidized ferredoxin and NAD[+][34], thus it is involved in electron bifurcation.

Margulisbacteria RA1A, all Saganbacteria and many Melainabacteria have genes encoding a putative butyryl-CoA

dehydrogenase (Bcd) in close proximity to genes encoding electron transfer flavoprotein subunits EtfA and EtfB (Fig. 3a). Together, these may form a Bcd/EtfAb complex, which is usually involved in electron bifurcation reactions between crotonyl-CoA, ferredoxin and NADH[35]. Specifically, Bcd catalyzes the transformation of short-chain acyl-CoA compounds to short-chain trans-2,3-dehydroacyl-CoA using electron-transfer flavoproteins as the electron acceptor (EtfAB). Nevertheless, the presence of an AMP-binding domain(s) instead of a second FAD-binding domain in the Etf indicates that electron bifurcation is not likely in this complex[36].

**Alternate mechanisms for the generation of a membrane potential.** Margulisbacteria AA1A, the selected representative genome for the oceanic Marinamargulisbacteria (82% completeness, Supplementary Data 1) shows signs of niche adaptation, including the ability to use $O_2$ as a terminal electron acceptor. This organism relies on aerobic respiration for generation of a $H^+$ potential, and instead of pyruvate fermentation to short-chain fatty acids or alcohols, it may use acetate as a source of acetyl-CoA (Supplementary Data 2). Phylogenetic analysis of the heme-copper oxygen reductase (complex IV) indicated that the potential for high-energy metabolism was an independently acquired trait (via HGT) in this organism (Fig. 7, Supplementary Fig. 5).

Like the sediment-associated Margulisbacteria, the oceanic Marinamargulisbacteria lack $CO_2$ fixation pathways. Thus, we anticipate that they adopt a heterotrophic lifestyle. The genome encodes all enzymes in the TCA cycle (including alpha-ketoglutarate dehydrogenase). NADH produced via glycolysis and the TCA cycle must be reoxidized. Unlike Riflemargulisbacteria that have membrane-bound NiFe hydrogenases, in Marinamargulisbacteria AA1A we identified genes encoding a six subunit $Na^+$-translocating NADH:ubiquinone oxidoreductase (Nqr; EC 1.6.5.8; *nqrABCDEF*; Supplementary Data 2). An electron transport chain in Marinamargulisbacteria AA1A (Fig. 3b) could involve electron transfer from NADH to the Nqr, from the Nqr to a menaquinone, then to a quinol:electron acceptor oxidoreductase (alternative Complex III; Fig. 5), and finally to a terminal oxidase (Fig. 7).

Phylogenetic analysis of the gene encoding the catalytic subunit (CoxA) of the terminal oxidase confirms that it is a type A heme-copper oxygen reductase (CoxABCD, EC 1.9.3.1; cytochrome *c* oxidase, (Fig. 7). The genes in these organisms are divergent and were probably acquired by horizontal gene transfer based on phylogenetic analysis (Supplementary Fig. 5). The recently released metagenome assembled Marinamargulisbacteria genome UBA6595 from Parks et al[13]. also encodes an alternative complex III and a type A heme-copper oxygen reductase. In contrast, the Riflemargulisbacteria do not have components of an ETC, except for a single gene of the cytochrome oxidase (*coxB*) presently with unknown function by itself (Supplementary Data 22). Interestingly, we did not identify any hydrogenases in the oceanic Marinamargulisbacteria, probably due to their different habitat, similar to many Cyanobacteria in the ocean[37].

As occurs in the oceanic Margulisbacteria, aerobic organisms within the Saganbacteria also rely on an ETC to generate a $H^+$ potential. Notably, based on phylogenetic analysis these aerobic organisms encode what looks like a novel type of heme-copper oxygen reductase (Fig. 7), and one organism (HO1A) also encodes the potential for anaerobic respiration (tetrathionate reductase; Fig. 5).

Saganbacteria HO1A and LO2A possess genes encoding a partial Nuo (complex I) that lacks the NADH binding subunits (NuoEFG) and could use ferredoxin as the electron donor instead

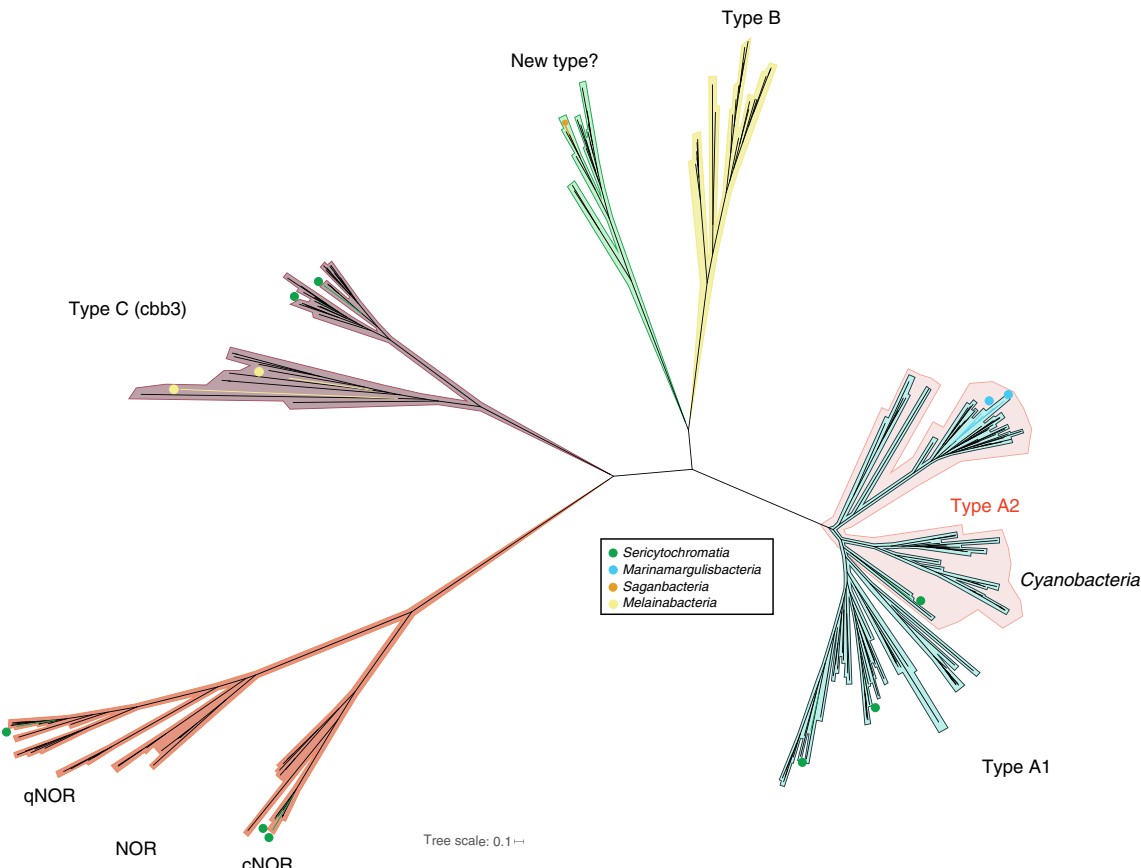

**Fig. 7** Phylogenetic tree of the catalytic subunit I of heme-copper oxygen reductases. The heme-copper oxygen reductase found in Saganbacteria seems to be a novel type based on phylogenetic analysis of catalytic subunits (CoxA). Marinamargulisbacteria have an A2 type heme-copper oxygen reductase, which is usually found in Cyanobacteria. The enzyme in Marinamargulisbacteria is part of a separate phylogenetic cluster from the aa₃-type heme-copper oxygen reductase found in Cyanobacteria (Supplementary Fig. 5). The heme-copper oxygen reductase found in Sericytochromatia LSPB_72[7] is type A1, and CBMW_12 has two enzymes[7]: one of type A1 and the other of type A2. Sericytochromatia LSPB_72 and CBMW_12 also have type C heme-copper oxygen reductases (CcoN)[7], like Melainabacteria in this study. Two types of bacterial nitric oxide reductase (NOR) have been identified in Sericytochromatia. One is a cytochrome *bc*-type complex (cNOR) that receives electrons from soluble redox protein donors, whereas the other type (qNOR) lacks the cytochrome *c* component and uses quinol as the electron donor. The tree is available with full bootstrap values in Newick format in Supplementary Data 25

(Supplementary Data 22). They also encode a succinate dehydrogenase (SDH) complex that is composed of four subunits and presumably fully functional as a complex II (Supplementary Data 2). Genes encoding a putative quinol:electron acceptor oxidoreductase complex, which has been previously seen in candidate phyla Zixibacteria[38], may transfer electrons to a terminal reductase and act as a complex III. Quinones, the usual electron donors for this and other respiratory enzymes, were not identified in this genome and alternative biosynthetic pathways should be investigated (Supplementary Data 2). Remarkably, these genomes also encode subunits of a quinol oxidase heme-copper oxygen reductase that seems to be a novel type, and that is closely related to a heme-copper oxygen reductase in candidatus *Methylomirabilis oxyfera* (NC10 phylum) (Fig. 7) of unknown type. It is unlikely that these organisms can reoxidize NADH via complex I, and it remains uncertain whether they can use O₂ as an electron acceptor. Downstream from the genes encoding the putative quinol reductase in Saganbacteria HO1A, there are genes encoding a tetrathionate reductase (Fig. 5), indicating that tetrathionate could be used as a terminal electron acceptor during anaerobic respiration, generating thiosulfate[39].

Melainabacteria include organisms capable of fermentation, respiration, or both[5,6]. In this study, Melainabacteria (except for AS1A and AS1B) have a partial complex I, similar to the one in Saganbacteria (Fig. 8, Supplementary Data 22). Melainabacteria, represented by genomes LO5B, HO7A and BJ4A, encode a partial SDH (complex II). Intriguingly, cytochrome $b_6$ (PetB; KEGG ortholog group K02635) and the Rieske FeS subunit (PetC; K02636) of the cytochrome $b_6f$ complex (complex III) were identified by HMMs predictions in these genomes. Only Melainabacteria LO5B, HO7A, and BJ4A are potentially able to use O₂ and other terminal electron acceptors. Specifically, Melainabacteria HO7A and BJ4A harbor O₂ reductases (see Supplementary Note 6) in the vicinity of genes encoding part of the cytochrome $b_6f$ complex (*petB* and *petC*), suggesting that they have a complex III/IV combination. Whether Melainabacteria are able to synthesize ubiquinone remains to be determined because only five out nine genes required for its synthesis were identified in the genomes of sediment-associated Melainabacteria (Supplementary Data 2).

Other forms of respiration were predicted in some Melainabacteria in this study. For instance, Melainabacteria LO5A encodes a cytoplasmic nitrite/nitrate oxidoreductase (NXR, Fig. 5). NXR participate in the second step of nitrification, the conversion of nitrite to nitrate and can also be reversible[40]. These genes are followed by another gene encoding cytochrome $b_6$ (*petB*) and a nitrate/nitrite transporter (*narK*; K02575).

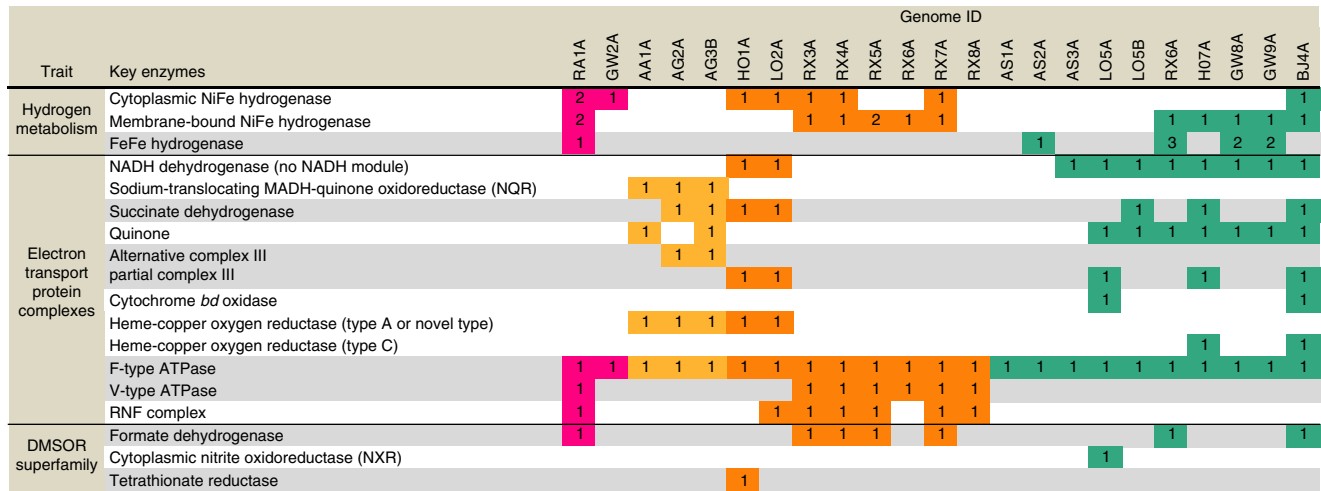

**Fig. 8** Key enzymes involved in H$_2$ and energy metabolism

## Discussion

Although billions of years have likely passed since the divergence of Cyanobacteria, Melainabacteria, Sericytochromatia, Margulisbacteria, and Saganbacteria, it is possible that features shared by the modern organisms were established prior to their divergence. Contemporary non-photosynthetic relatives of the Cyanobacteria share several interesting genomic features that frame their energy metabolism. These include numerous hydrogenases, lack of an oxidative phosphorylation pathway in many organisms, widespread lack of carbon dioxide fixation pathways, and the presence of multiple types of membrane-bound complexes that may be involved in ion translocation. Against this background, oxygenic photosynthesis and aerobic respiration represent a dramatic net-energy gain and access to a different life strategy – one that does not rely on limited organic carbon as a source of electrons, or complexes that provide only low-energy yields linked to ion translocation. Based on our analyses, we suggest that the common ancestor of all of these organisms was more likely an anaerobe with hydrogen-based, fermentation-based metabolism than the alternative scenario, in which the ancestor was an aerobe and all but Cyanobacteria subsequently lost this capacity. Even today, Cyanobacteria can switch to fermentative metabolism under anoxic conditions, and at the heart of this process is a bidirectional group 3d NiFe hydrogenase[41]. Furthermore, the presence of a low complexity nitrogenase operon in Riflemargulisbacteria suggests that this enzyme may have been present in anaerobic bacteria before the split between Cyanobacteria and Riflemargulisbacteria.

Anaerobic representatives of Riflemargulisbacteria and Saganbacteria have multiple protein complexes that may participate in the coupling of cytoplasmic redox reactions to ion translocation across the membrane. In Riflemargulisbacteria and Saganbacteria, membrane-bound protein complexes (e.g., group 4 NiFe hydrogenases and hydrogenase-related complexes) could make up a minimal, highly efficient respiratory chain, as previously suggested for these types of enzymes[19]. Sequential translocation of ions across membranes is required in fermentation-based metabolisms as the energy yield per translocation is low[15]. Riflemargulisbacteria is unique among the anaerobic lineages because its genome is predicted to encode multiple cytoplasmic H$_2$-consuming hydrogenases, in addition to membrane-bound energy-converting H$_2$-evolving NiFe hydrogenase (Ech), a Rnf complex, and potentially an FDH-Ehr complex. The cytoplasmic hydrogenases constitute the main mechanism by which these bacteria oxidize reducing equivalents that result from core metabolic

pathways and produce H$_2$. The main function of the Ech hydrogenase and the Rnf complex may be to couple ferredoxin oxidation or other kinds of cytoplasmic redox reactions with ion-translocation;[21,24,42,43] and a V-type ATPase could be involved in ATP synthesis from the ion potential generated from both these complexes. Anaerobic Saganbacteria that also possess a Rnf complex and a V-type ATPase may be able to use this mechanism for energy conservation as well. However, the Rnf complex is reversible (like the Ech hydrogenase) and may play a role in substrate transport instead of ATP synthesis[24].

Two groups of closely related Saganbacteria have group 4 hydrogenases with unknown electron donors that may also be involved in the creation of a membrane potential. Saganbacteria like Riflemargulisbacteria, could interconvert energy between an electrical gradient and an ionic gradient via group 4 NiFe hydrogenases. Given that the membrane-bound NiFe hydrogenases in Saganbacteria RX5A and RX6A have antiporter-like subunits, we suspect that they may function like Mbh hydrogenases in other organisms. For instance, in *Thermococcus onnurineus*, an Mbh hydrogenase produces H$_2$ from the oxidation of reduced ferredoxin, creating a H$^+$ potential across the membrane that is converted to a secondary Na$^+$ gradient by the antiporter subunits[44,45]. The energy stored in electrochemical gradients can be used by ATPases to synthesize ATP.

The possible FDH-Ehr complex in anaerobic Riflemargulisbacteria and Saganbacteria is unusual, and may be involved in oxidative phosphorylation. In this scenario, the FDH-Ehr complex would serve as a simplified electron transport chain. We suspect that in Riflemargulisbacteria and Saganbacteria the Nuo-like electron transfer subunits encoded next to the catalytic subunit of formate dehydrogenase (FdhA) are possibly part of a multimeric formate dehydrogenase (FDH). The formate dehydrogenase together with the hydrogenase-like subunits in the Ehr may pass electrons down to the transmembrane subunits, where an electron carrier such as quinone could be reduced and ion translocation would take place. For other Ehr complexes (e.g., Mbx), it was suggested that the distance between the formate dehydrogenase and membrane-bound subunits is important for the creation of an electrochemical potential across the membrane[16].

Experimental testing is needed to determine whether these Ehr complexes are part of a complex with formate dehydrogenase or another oxidoreductase, are expressed, assembled, and functional. The existence of an FDH-Ehr complex is even more questionable in Melainabacteria BJ4A, because the genes encoding the electron

transfer subunits in the vicinity of *fdhA* were not identified. Such complexes were not found in previously described Melainabacteria either[5]. Regardless of the true function of the putative FDH-Ehr complexes, it is interesting to discover the variety of other membrane-bound energy-conserving complexes in lineages sibling to Cyanobacteria. Membrane-bound group 4 NiFe hydrogenases and related complexes share a common ancestor with Nuo, in which the catalytic subunits of NiFe hydrogenases may have later become the electron transfer subunits of complex I[16,17,46]. Nuo couples NADH oxidation with $H^+$ or ion translocation[16] and is a key component of respiratory and photosynthetic electron transport chains in other organisms[17].

Notable in Riflemargulisbacteria in particular (but present in other groups) are a variety of potential electron bifurcating complexes, including Nfn[47]. Although not described as such, cytoplasmic NiFe hydrogenases (groups 3c) and cytoplasmic bidirectional FeFe hydrogenases could be involved in electron bifurcation as well. Electron bifurcating complexes are common in fermentative organisms, and those found in Riflemargulisbacteria probably serve as the primary mechanism to balance redox carriers in these organisms.

From the perspective of metabolic evolution, it is interesting to consider the origin of high electrode potential electron transport chains and their components. In this study, we were particularly interested in the electron entry (complex I) and exit points (terminal oxidases). Within the Saganbacteria and Melainabacteria, Saganbacteria HO1A and LO2A and Melainabacteria LO5B, HO7A and BJ4A are the only organisms that possess some sort of an electron transport chain. Given the prediction that the common ancestor of all groups was a fermentative anaerobe, we suspect that the ETC and ability to use $O_2$ was a later acquired trait.

Saganbacteria HO1A and LO2A and most Melainabacteria in this study (except AS1A and AS2A) have in common with Cyanobacteria the lack of the NADH module of NADH dehydrogenase (complex I). Respiration in the thylakoid membrane of Cyanobacteria is mostly initiated from succinate rather than NADH. However, how succinate dehydrogenase (SDH; complex II) works in *Synechocystis* sp. is not completely understood, in part due to the lack of two (out of four) subunits[48,49]. This is also the case in Melainabacteria LO5B, HO7A and BJ4A. The complex I found in Saganbacteria HO1A and LO2A and Melainabacteria may be involved in proton translocation independent of NADH oxidation, and thus it may play a role similar to that of membrane-bound NiFe hydrogenases. Remarkably, Saganbacteria HO1A and LO2A also carry a putative novel type of heme-copper oxygen reductase (quinol oxidase), which could act as the terminal reductase in an electron transport chain.

Only Marinamargulisbacteria is predicted to have an electron transport chain that includes an alternative complex III (ACIII) and a type A heme-copper oxygen reductase (cytochrome oxidase adapted to high $O_2$ levels), which may act as the intermediary between complex I and the terminal oxidase. Marinamargulisbacteria UBA6595[13] and Sericytochromatia CBMW_12[7] are the only other related bacteria known to have an ACIII and a type A heme-copper oxygen reductase. Phylogenetic analyses indicate that Marinamargulisbacteria like Sericytochromatia[7] acquired this complex by lateral gene transfer. Given that Marinamargulisbacteria AA1A was found in the ocean, where dissolved $O_2$ may be available, it must have acquired the necessary machinery (*i.e.*, complex I, menaquinone, ACIII, cytochrome oxidase) to take advantage of $O_2$ as a terminal electron acceptor. Similarly, fermentation and $H_2$ metabolism may be less relevant in the water column, and this clade may have lost the ancestral traits that were advantageous in other redox conditions.

The type C heme-copper oxygen reductase (complex IV) found in Melainabacteria HO7A and BJ4A may be adapted to low $O_2$ levels, as expected for microaerophilic or anoxic environments such as the subsurface and the human gut. Some aerobic Melainabacteria studied here, as well as *Obscuribacter phosphatis*[6,7], have a fusion involving complex III and complex IV. However, we annotated the proteins related to complex III in these Melainabacteria as cytochrome $b_6f$ subunits. If correct, this is important because complex $b_6f$ has only been found to play the role of complex III in Cyanobacteria[7]. Thus, cytochrome $b_6f$ complex may have been present in the common ancestor of all of these lineages.

Notably, Melainabacteria BJ4A could have a branched electron transport chain, with one branch leading to a cytochrome *d* ubiquinol oxidoreductase and the other leading to the type C heme-copper oxygen reductase. When two $O_2$ reductases are present in other lineages they tend to have different $O_2$ affinities. For instance, Cyanobacteria have a type A heme-copper oxygen reductase in the thylakoid membrane and a quinol cytochrome *bd* oxidase in the cytoplasmic membrane (in addition to a cytochrome *bo* oxidase)[48]. Sericytochromatia also have two types of heme-copper oxygen reductases, type A and type C, that may also differ in their affinity for $O_2$[7].

Based on the absence of genes for $CO_2$ fixation and photosynthetic machinery in Melainabacteria and Sericytochromatia, the lineages most closely related to Cyanobacteria, it was suggested that these capacities arose after their divergence[2,5,7]. However, the alternative possibility is that they were a characteristic of their common ancestor, but lost in Melainabacteria and Sericytochromatia. The current analyses support the former conclusion, given the essentially complete lack of genes that may be involved in carbon fixation and photosynthesis in two additional lineages sibling to Melainabacteria, Sericytochromatia, and Cyanobacteria.

There is considerable interest in the metabolism of lineages sibling to the Cyanobacteria. Genomes from these lineages may provide clues to the origins of complexes that could have evolved to enable aerobic (and other types of) respiration via an electron transport chain. Based on the analyses presented here, we suggest that $H_2$ was central to the overall metabolism, and hydrogenases and the Rnf complex played central roles in proton translocation for energy generation, with redox carrier balance reliant upon electron bifurcation complexes.

## Methods

**Samples collection, DNA extraction, and sequencing.** Publicly available genomes in this study were recovered from several sources (Supplementary Data 1). Margulisbacteria GW1B-GW1D, Saganbacteria HO1A, HO1B, LO2A, LO2B, HO2C, RX3A-RX8C, Melainabacteria HO7A, LO5A, LO5B, RX6A-RX6C, GW8A, and GW9A originated from an alluvial aquifer in Rifle, CO, USA (groundwater samples; see Supplementary References). For sampling, DNA processing, sequencing information, metagenome assembly, genome binning and curation of publicly available and newly generated genomes see Supplementary Methods.

**Data processing, assembly, binning, and curation of newly generated MAGs.** For description of reads trimming, assembly algorithm, binning methods and genome curation see Supplementary Methods.

**Completeness estimation.** Genome completeness was evaluated based on a set of 51 single copy genes previously used[8]. Genomes from metagenomes and single-cell genomes were categorized according to the minimum information about a metagenome-assembled genome (MIMAG) and a single amplified genome (MISAG) of bacteria and archaea[12].

**Functional annotations.** Predicted genes were annotated using Prodigal[50], and similarity searches were conducted using BLAST against UniProtKB and UniRef100[51], and the Kyoto Encyclopedia of Genes and Genomes (KEGG)[52] and uploaded to ggKbase (http://www.ggkbase.berkeley.edu). Additionally, the gene

products were scanned with hmmsearch[53] using an in house HMMs database representative of KEGG orthologous groups. More targeted HMMs were also used to confirm the annotation of genes encoding hydrogenases[8]. Furthermore, protein domains were predicted using Interpro[54] and CD search[55].

**Gene content comparison**. Inference of clusters of orthologous proteins was performed with OrthoFinder 2.1.3[56] on a set of genomes representing the following class and phylum level lineage-specific pangenomes: Margulisbacteria ($n = 14$), Saganbacteria ($n = 26$), Sericytochromatia ($n = 3$), Melainabacteria ($n = 11$), and Cyanobacteria ($n = 34$). Pangenomes include all proteins found in the respective set of genomes. Cyanobacterial genomes were selected based on IMG taxonomy strings. For each cyanobacterial genus, one genome with a predicted contamination <5 % (determined with CheckM[57]) was included in the dataset (Supplementary Data 23). For each class/phylum-level lineage all proteomes of members of the respective lineage were combined to pan-proteomes. Protein families were identified with OrthoFinder[56]. Protein families that contained only proteins from a single lineage were treated as singletons and excluded from the following analysis. For all possible lineage pairs the percentage of shared protein families was calculated as the total number of protein families the respective lineage shared with the other divided by the total number of protein families found in this lineage. As lineage pan-proteomes differed in size, the average of the bidirectional percentage of shared protein families was taken for each pair. The pangenome comparisons were visualized using the python packages matplotlib and pyUpSet (https://github.com/ImSoErgodic/py-upset).

**Phylogenomics**. Two different species trees were constructed, the first tree to place Saganbacteria and Margulisbacteria into phylogenetic context with other bacterial phyla and the second one to provide a more detailed view of taxonomic placement of different Saganbacteria and Margulisbacteria. To reduce redundancy in the first species tree DNA directed RNA polymerase beta subunit 160kD (COG0086) was identified in reference proteomes, Saganbacteria and Margulisbacteria using hmmsearch (hmmer 3.1b2, http://hmmer.org/) and the HMM of COG0086[58]. Protein hits were then extracted and clustered with cd-hit[59]. A de-replicated set of reference genomes was obtained from publicly available bacterial genomes in IMG/M[60] by COG0086 clustering at 65% sequence similarity and further de-replication of overrepresented clades. Cluster-representatives with the greatest number of different conserved marker genes were used to build the species tree of the Terrabacteria (Supplementary Data 24). For the detailed species tree (Fig. 1b), pairwise genomic average nucleotide identity (gANI) was calculated using fastANI[61]. Only genome pairs with an alignment fraction of >70% and ANI of at least 98.5% were taken into account for clustering with MCL[62]. Representatives of 98.5% similarity ANI clusters were used for tree building based on two different sets of phylogenetic markers; a set of 56 universal single copy marker proteins[63,64] and 16 ribosomal proteins[14]. For every protein, alignments were built with MAFFT (v7.294b)[65] using the local pair option (mafft-linsi) and subsequently trimmed with BMGE using BLOSUM30[66]. Query genomes lacking a substantial proportion of marker proteins (<28 out of 56) or which had additional copies of more than three single-copy markers were removed from the data set. Single protein alignments were then concatenated resulting in an alignment of 13,849 sites for the set of 56 universal single copy marker proteins and 2143 sites for the 16 ribosomal proteins. Maximum likelihood phylogenies were inferred with IQ-tree (multicore v1.5.5)[67] using LG + F + I + G4 as suggested (BIC criterion) after employing model test implemented in IQ-tree. To milder effects of potential compositional bias in the dataset and long-branch attraction in the 56 universal single copy marker protein trees, the concatenated alignments were re-coded in Dayhoff-4 categories[68–70] and phylogenetic trees were calculated with PhyloBayesMPI[68] CAT + GTR in two chains, which both converged with maxdiff = 0.09 for the Terrabacteria species tree and maxdiff = 0.11 for the detailed species tree. The first 25% of trees in each chain were discarded as burn-in. Phylogenetic tree visualization and annotation was performed with ete3[71].

**Catalytic subunit of NiFe and FeFe hydrogenases phylogenetic tree**. Based on existing annotations target proteins were identified in query proteomes and reference organisms. Identical sequences were removed from the data set, alignments built with MAFFT (v7.294b)[65], trimmed with trimal[72] (removal of positions with more than 90% of gaps) and maximum-likelihood phylogenetic trees inferred with IQ-tree (multicore v1.6.6)[67] and the best fit model based on model test in IQ-tree (including mixture models LG4M, LG4X, C20, C40, and C60) and Bayesian phylogenetic trees inferred with PhyloBayesMPI (version 1.7)[68]. Phylogenetic models used for the final trees were C40 + R7 in IQ-tree for the FeFe Hydrogenases and PhyloBayesMPI CAT + GTR (version 1.7)[68] in two chains for the NiFe Hydrogenases, which both converged with maxdiff of 0.25 (Groups 1, 2, 3 NiFe hydrogenases), and 0.22 (Group 4 NiFe hydrogenases and related complexes). The first ~30% of trees were discarded as burn-in. Phylogenetic trees were visualized in ete3[71] and iTol[73].

**NifHDK phylogenetic tree**. HMMs for NifH, NifD, and NifK were downloaded from TIGRFAM[74] and used to identify NifHDK in ~70,000 microbial genomes in IMG/M[60] using hmmsearch (hmmer 3.1b2, http://hmmer.org/). Significant hits for NifHDK were extracted from genomes which encoded all three genes. Sequences

were aligned with mafft[65], de-replicated by clustering with cd-hit[59] at a similarity cutoff of 90%, and HMMs were built using hmmbuild (hmmer 3.1b2, http://hmmer.org/). The improved HMMs were then used to identify NifHDK in novel genomes and microbial genomes available in the IMG/M system using hmmsearch (hmmer 3.1b2, http://hmmer.org/). Protein hits were extracted and de-replicated based on clustering of NifK at 90% sequence similarity with MCL[62]. NifHDK of cluster medoids were then aligned with mafft[65], and alignments trimmed with trimal[72] to remove positions with more than 90% gaps. A phylogenetic tree was built on a concatenated alignment of NifHDK with IQ-tree LG4M + R10 as suggested (BIC criterion, including mixture models LG4M, LG4X, C20, C40, and C60) in IQ-tree (multicore v1.5.5)[67].

**Catalytic subunit of dimethyl sulfoxide reductase superfamily protein and catalytic subunit I of heme-copper oxygen reductases phylogenetic trees**. Each individual protein data set was aligned using Muscle version 3.8.31[75,76] and then manually curated to remove end gaps. Phylogenies were conducted using RAxML-HPC BlackBox[77] as implemented on the CIPRES web server[78] under the PROTGAMMA JTT evolutionary model and with the number of bootstraps automatically determined.

**HCO A-family oxygen reductase protein tree**. Homologs of CoxA in Marinamargulisbacteria and Saganbacteria genomes were identified from HMM searches. Other homologs were gathered from Shih et al.[2]. Sequences were aligned with MAFFT using the –maxiterate[79]. Phylogenetic analysis was performed using RAxML through the CIPRES Science Gateway[77] under the LG model.

**Statement of ethics**. The fecal samples obtained were part of a clinical PhaseI/II study in rural Bangladesh entitled "Selenium and arsenic pharmacodynamics" (SEASP) run by Graham George (University of Saskatchewan) and. The SEASP trial was approved by the University of Saskatchewan Research Ethics Board (14-284) and the Bangladesh Medical Research Council (940,BMRC/NREC/2010-2013/291). Additional ethics approval was also obtained by UCL (7591/001). The study complied with all the relevant ethical regulations. Informed consent was obtained from all human participants.

**Reporting Summary**. Further information on experimental design is available in the Nature Research Reporting Summary linked to this article.

## Data availability

New and published genomes included in this study and corresponding gene annotations can be accessed at https://ggkbase.berkeley.edu/Margulis_Sagan_Melaina/organisms (ggKbase is a 'live' site, genomes may be updated after publication). DNA sequences (new genomes and raw sequence reads) have been deposited in the NCBI Bioproject Database (accession codes: PRJNA167727, PRJNA451230, PRJNA471730, PRJNA471718). Further details are provided in Supplementary Data 1, including NCBI Genbank accession numbers for individual genomes. A reporting summary for this Article is available as a Supplementary Information file.

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

## Acknowledgements

Laura Hug and Christopher Brown provided assistance with analyses, Shufei Lei and Kate Lane assisted with bioinformatics and data management, Kelly Wrighton, Kim Handley, and Kenneth Hurst Williams provided samples. Sallie Chisholm, Paul Berube, and Steven Biller are acknowledged for their help securing the marine samples. We thank the staff of Bigelow Laboratory Single-Cell Genomics Center for the generation of single-cell data. Teruki Iwatsuki, Kazuki Hayashida, Toshihiro Kato, and Mitsuru Kubota assisted with groundwater sampling at Mizunami Underground Research Laboratory, Japan Atomic Energy Agency (JAEA). Thanks to Chris Greening for feedback regarding FDH-Ehr complexes on the bioRxiv version of this manuscript, and to Denis Baurain and an anonymous reviewer for all the useful comments on the manuscript. The research was supported by the Department of Energy (DOE), Office of Science and Office of Biological and Environmental Research (Lawrence Berkeley National Lab; Operated by the University of California, Berkeley). DNA sequencing for the Rifle samples was conducted by the U.S. Department of Energy Joint Genome Institute, a DOE Office of Science User Facility, supported under Contract No. DE-AC02-05CH11231. Marine single amplified genomes were generated and sequenced with the support of NSF grants DEB-1441717 and OCE-1335810, and Simons Foundation grant 510023 (to R.S.). Fecal samples were collected from patients in the clinical PhaseI/II SEASP trial in Bangladesh that was jointly led by Graham George and Ingrid Pickering (University of Saskatchewan), with the assistance of the SEASP team https://clinicaltrials.gov/ct2/show/NCT02377635, and funded by the Canadian Federal Government, through Grand Challenges Canada, Stars in Global Health and by the Global Institute for Water Security. The study was funded by the Canadian Federal Government, through a program entitled Grand Challenges Canada, Stars in Global Health, with additional funds from the Global Institute for Water Security at the University of Saskatchewan.

## Author contributions

P.B.M.C., I.S., F.S., B.C.T., M.R.O., and J.F.B. reconstructed and curated the genomes; P. B.M.C. conducted the majority of the metabolic analyses, F.S., C.J.C., R.K., D.B., P.S., K. A., and J.F.B. provided input to analyses; F.S., C.J.C., P.S., and P.B.M.C. generated phylogenetic trees; Y.A., E.D.B., and R.S. acquired samples; Y.A., J.M.S., E.D.B., and R.S. acquired new sequence information; and E.D.B., R.S., and T.W. contributed single amplified genomes (SAGs). P.B.M.C. and J.F.B. wrote the manuscript with input from F. S. and T.W., as well as C.J.C., P.S., and R.K.; All authors reviewed the results and approved the manuscript.

## Additional information

**Competing interests:** The authors declare no competing interests.

