## [Peer Review File · Nature Communications]

Reviewers' comments:

Reviewer #1 (Remarks to the Author):

Caveat: I am a phylogenomicist with first-hand experience in bacterial metagenomics, cyanobacterial phylogenetics and microalgal respiratory and photosynthetic metabolisms. In contrast, I am not an expert in all the peculiarities of the bacterial metabolisms. Therefore, the extensive metabolic discussions found in this manuscript were a tough read for me, and even if the authors appear to be very knowledgeable, I am afraid that this article might be difficult to follow for the broad readership of Nature Communications. Maybe the authors could consider expanding their introduction and adding a more streamlined overview of their results to help the non-experts reader to figure out what to recall from their study (see below).

Denis Baurain
University of Liège

General comments:

In this manuscript, Carnevali and co-workers mined new and existing metagenome and single-cell genomes from bacterial phyla supposedly related to Cyanobacteria in order to infer the ancestral metabolism of the whole group. They first used two small phylogenomic supermatrices to confirm that the phyla under study are indeed related to Cyanobacteria and how they relate to each other, and then searched for genes involved in energetic metabolism, building single-gene (family) trees to support their sub-classification of the various genes recovered.

Generally speaking, the work is thorough and appears to be methodologically sound. The figures were prepared with care and are graphically appealing. Therefore, I do not think that any "experiment" should be redone.

My main concern is the legibility of the whole manuscript, which is quite complicated to follow for someone not highly acquainted with non-conventional energetic metabolism. In particular, the three first introductory paragraphs are more confusing than anything, e.g., mixing ETCs for respiration and photosynthesis, assimilating respiration to O₂ use and fermentation to a fallback for anaerobic conditions. Here are a few background questions that might help the authors to better introduce the metabolic concepts discussed in their study. What is energy conservation and what is high-energy metabolism? What is the difference between respiration and fermentation? What is anaerobic respiration? Are there anaerobic ETCs? Can fermentation and anaerobic respiration cohabit in a single organism? What are alternative respiratory complexes? What are electron-bifurcating complexes and how do they achieve energy conservation? What are the various electron donors and acceptors that can be found in Bacteria? What is H₂ metabolism and how is it articulated with fermentation and anaerobic respiration?

Specific comments:

Abstract

* Two sentences are a too vague to be really useful: [lines 60-61: genomic features suggest the capacity for metabolic fine-tuning under strictly anoxic conditions] and [69-71: Oxyphotobacteria may have acquired the ability to use O₂ as a terminal electron acceptor under certain environmental conditions] Please try to be more specific, especially for the last sentence.

Introduction

* Given the age of the GOE and the very wide distribution of respiratory systems, I am not convinced that any inference from modern organisms can really "constrain the metabolic platform into which complex ETCs evolved" [lines 104-107]. For example, Melainobacteria were first described from human gut microbiome, which is certainly not the original environment where they

have evolved from their common ancestor with the other lineages of interest. Thus, I would like the authors to be more cautious when explaining how their observations of modern organisms can help determining the lifestyles of their common ancestors [lines 119-122].

Results

* All this study rests upon the accuracy of the reconstruction of the metagenomes under study.

Apparently, 4 Saganbacteria genomes are circularized [lines 133-135], which suggests that they are completely assembled and thus more susceptible to give a faithful picture of their gene content. What about the other genomes? For example, the co-occurrence of Ech hydrogenase and Rnf complex is deemed to be rare yet observed in one Riflemargulisbacteria (RA1A) [lines 220-222]. Similarly, the lone *coxB* in the same Riflemargulisbacteria is puzzling [lines 278-280]. A sentence giving the limits of the study in terms of binning accuracy would be useful here.

* From the Results only, it is not clear if the phylogenomic analyses are new or not [line 153-156]. Methods are detailed but references are given in the Results. This should be clarified. Moreover, are the two datasets independent? Are we sure that the 56 universal single copy proteins do not include any of the 16 ribosomal proteins? Otherwise, introduce the second as a subset of the first. Regarding phylogenetic resolution, some nodes have no symbol in Figure 1a-1b (and many other figures); do they correspond to maximal support? If so, this should be stated in the figure legends.

* The analysis of shared protein families is laudable but suffers from some caveats that should be explained. OrthoFinder always creates hundreds of thousands of gene "families", ranging from very large groups composed of dozens of proteins per genome to (numerous) singletons containing only one single protein from one single genome. From Figure 1c, it is clear that "family" is used in that technical meaning, since Oxyphotobacteria alone (34 genomes) have 120,000 families! In my opinion, singletons should be removed from the pangenome statistics (or at least counted separately). By the way, how (and which) were the representative cyanobacterial genomes selected [lines 698-699]? Some publicly available cyanobacterial genomes are quite contaminated by foreign DNA (and proteins) and this might have affected the conclusions by adding a background of non-cyanobacterial families [Cornet et al. 2018, PLOS ONE; <https://www.biorxiv.org/content/early/2018/04/15/301788>]. Regarding the way shared families are computed, I don't understand the details [lines 701-704]. What is 100% in Figure 1c? When does a family count as "shared"? As soon as one protein from one of the genomes composing the pangenome is found in the same OrthoFinder group as another protein from another pangenome? Finally, the incongruence between the phylogenetic relationships deduced from the supermatrices and the gene content similarities computed between pangenomes [lines 166-171] argue for being very cautious when inferring the gene content and the metabolism of the common ancestor (see comment above).

* Some protein complexes are discussed using different names. This makes life harder for the casual reader. Please try to be consistent throughout the text and the figures (e.g., NiFe hydrogenases from group 4f = putative = Ehr?; group 4e = Ech?)

* The discussion about the architecture of V-type ATPase and Rnf operons [lines 222-229] suggests that HGT may have played a role in the emergence of ETCs. If so, should it be discussed here and does it affect inferences about ancestral metabolism?

Discussion

* The first paragraph [lines 408-414] suffers from the same interpretational issues as those pointed out in the introduction. It is too affirmative in its inference of the ancestral metabolic background in which have emerged photosynthesis and aerobic respiration, considering the age of the GOE and the subsequent niche adaptations observed by the authors in the sister lineages of Oxyphotobacteria. I am not saying that the authors are wrong, but I would like them to better differentiate between results in contemporaneous organisms and inferences about the past.

* Maybe more provocatively, in terms of significance, what could have been the common ancestor of all these phyla other than what is hypothesized here, i.e., an anaerobe with a fermentation-based metabolism, before the advent of photosynthesis and aerobic respiration [lines 419-423]? The alternative is not clear to me... but this is likely due to my own ignorance on the bacterial

metabolisms.

* The last sentence of the discussion [553-555] is not informative. It might be better to reformulate it to make it less obvious.

Methods

* Is it correct that Margulisbacteria RA1A is one bin out of 361 bins assembled from 14 samples [lines 607-618]? If so, there is ample room for binning errors, as suggested above. Please, add a sentence about this when introducing such metagenomes. If not, clarify the origin of this metagenome.

* Still about this genome, I fail to see how the mini-assembly procedure has managed to reduce the number of scaffolds from 147 to 111 without adding longer reads [lines 627-638]. Is it because the various assembly thresholds are somewhat lower during that procedure than during the original assembly with idba_ud?

* How exactly was performed the phylogenetic affiliation of the contigs [lines 647-648]? What database could provide enough taxonomically reliable reference sequences for that task? How much contigs had to be removed (from the 111?) after that control?

* Gene content comparison is not enough described (see comment above). Moreover, I could not find the list of the selected cyanobacterial genomes [lines 694-705].

* The selection of Terrabacteria genomes for the phylogenomic tree is not that clear to me. I understand that genomes (?) were dereplicated based on RNA polymerase [lines 711-715]. But how large and from where was the non-reduced set of genomes? Besides, is it correct that representative genomes were selected based on the fact that their RNA pol had been retained as the cluster representative by cd-hit? If so, this is not that smart given that (i) the only criterion to be the cd-hit representative is being the longest (first seen) sequence of the cluster and that (ii) fast-evolving organisms (not the best choice for phylogenomic trees) often have lengthened protein sequences. Please clarify here. More generally, this part of the Methods is difficult to follow in terms of datasets and figures (see also comment below about Supplemental Figures).

* The authors are very honest in their description of the various single-gene tree-building strategies [lines 736-762]. This is laudable. However, I fail to see why methods vary so widely between the different trees: different species sampling strategies, different aligners, different phylogenetic inference packages and evolutionary models. Could this be justified or is it just the result of many authors working each on their own?

Wording and typos:

* line 108 (and others): "sibling" is better written as "sister" in a phylogenetic context

* line 119: "aerobically" is written twice

* line 219: Fig. 2a should be Fig. 3a

* line 247: nincotinamide => nicotinamide

* line 287: Saganbacteria "are" heterotrophic anaerobes should be "include" considering the previous sentence about both aerobic and anaerobic representatives

* line 300: FHL should be spelled out in full before line 446

* line 446: "the" is written twice

* line 481: the end of the sentence has an odd grammar

* line 490: posses => possess

* line 860: there is an extra "or" before NifJ

* line 868: define "close proximity"

* line 869: puple => purple

* line 917: (in the table) Trasnport => Transport

Figures:

* Figure 3a: it might be useful to add the "e" and "f" to the corresponding group 4 NiFe hydrogenases for clarity. Moreover, there are two asterisks (*) that are not explained in the legend.

- * Figure 3b: why not to show H⁺/Na⁺ translocation explicitly?
- * Supplemental Figure 1b is identical to Supplemental Figure 2. I am pretty sure that this is the result of a cut-and-paste error.
- * Supplemental Figure 6: the zoomed part would gain in legibility by annotating the various phyla observed in the tree.

Reviewer #2 (Remarks to the Author):

The paper by Metheus Carnevali et al explores the functional composition of MAGs or SCGs affiliated with groups that branch sister to Cyanobacteria. The authors characterize the putative metabolisms of MAGs/SCGs to provide insight into the metabolism of the ancestor of Cyanobacteria. Results are used to suggest that the ancestor is a fermentative bacterium that likely relies on group 4 NiFe hydrogenases or RNF complexes to drive ion translocation and generate ATP. The paper is of general interest but suffers from poor organization and overinterpretation in places. The following list of major and minor comments may help to improve the paper.

The title is potentially misleading. I don't think it is common to call fermentative microorganisms hydrogen based. They are carbon based and generate H₂.

Roughly a quarter of microbial genomes encode for one or more hydrogenases and 75% of the genomes of anaerobes encode bifurcating enzymes. I don't see these characteristics as being particularly unique to these organisms, siblings of Cyanobacteria or not.

The paper generally suffers from difficulty in its organization and the use of acronyms or loosely defined definitions. For example, is Oxyphotobacteria really a replacement for Cyanobacteria and has this been widely accepted in the field. I see this referenced out of Fischers group where it is claimed that this group postdates the rise of O₂ – not sure how this could be making me ever more skeptical of this new name. Moreover, how does this term capture anoxygenic photosynthesis by Cyanobacteria, or even fermentative metabolism within Cyanos that the authors reference in their paper.

The authors discussion of nitrogenase could be improved, and if done correctly, extremely revealing. NifH is a poor predictor of the phylogeny of Mo-nitrogenases. This needs to be done with a concatenation of NifHDK. Two monophyletic lineages will emerge in that phylogenetic reconstruction, those NifHDK hosted by aerobes/facultative anaerobes and those from strict anaerobes. It would be interesting to see where the NifHDK from this putative anaerobe and potential ancestral state of Cyanobacteria branch.

In line 81, the authors present a paradox – oxygenic phototrophs (Cyanobacteria) require O₂ to drive the biosynthesis of some of the components of their cellular machinery. Yet, this is written as though the only opportunities to take advantage of O₂ were made available after they evolved. So, what came first – the ability to drive O₂ formation or the ability to synthesize components that require O₂? Perhaps all of this occurred stepwise in some logical way from a fermentative ancestor? Lots would need to take place including acquisition of CO₂ fixation and chlorophyll biosynthesis componentry.

Minor comments

Line 78: There would have been O₂ prior to the GOE but it would not have accumulated in the atmosphere. It would have likely been available for use by microbes, however, unless the kinetics of abiotic reactions were just that fast.

Line 79: No respiratory processes prior to O₂ then? What about Fe reduction or the Mtr complex in methanogens?

Line 98: Energy is not conserved during the reduction of O₂, but rather from the conversion of reducing equivalents to an electrochemical potential during ion translocation.

Line 181: Sodium

Lines 182: From where and how recently? This is actually critical to your paper – perhaps the ancestor was like a cyanobacterium and diverged to form these organisms comparatively recently by acquisition of a number of anaerobic processes.

Line 188: All life uses Fd. There are just different forms with different potentials that allow Fd use in aerobes.

Line 202: The discovery of FeFe hydrogenase in lineages sister to Cyanobacteria and the suggestion of ancestral character is an overlooked point here. It has long been problematic to the hydrogenase community why Cyanobacteria only encode NiFe hydrogenase while the rest of bacteria encode FeFe and NiFe hydrogenases. The other group of oxygenic phototrophs, algae, use FeFe hydrogenases but not NiFe hydrogenases. If the authors are correct in the overall interpretation of this paper, it would suggest that FeFe hydrogenases were lost and NiFe were retained. This may have to do with the FeFe being bifurcating type (see caveat below) and these are never found in the genomes of aerobes (they won't work).

Line 214: Converting.

Line 216: These can be in numerous cells. The Greening paper artificially breaks these up by taxonomy (rather than function) in their phylogenetic assignments.

Line 221: Just a comment – the presence of Ech and Rnf, plus numerous bifurcating complexes, really does suggest these organisms are living on the thermodynamic edge. Wonder if this was the trigger to evolve something more splendid metabolically? i.e., oxygenic photosynthesis.

Line 232: These are not NiFe hydrogenases if they lack the distal and vicinal Cys pairs that ligate the NiFe active site cluster. See Schut et al., 2013 and Schut et al., 2016 for a quick primer on what they could be.

Line 243: A reference is needed for this statement

Line 245: The authors need to do more than to place the catalytic subunit of FeFe hydrogenase (HydA) functionally than just phylogenetics ala Greening 2016. If it is bifurcating, it will have HydBC (possibly HydD) in the flanking gene environment (see Poudel et al., 2016) for a classification scheme that is accurate.

Line 250: Thieren et al., 2017 have suggested FeFe hydrogenase variants that are regulated by N availability and that are catalytically biased towards H₂ oxidation.

Line 272 and elsewhere: Phylogenetic analyses do not confirm that this is a type A heme-copper oxygen reductase adapted to atmospheric O₂ levels. This type of overinterpretation through phylogenetics and gene/inferred protein homology detracts from the paper and is unnecessary.

Line 281: Are they common in marine Cyanobacteria? I believe so.

Line 296: These are not sulfhydrogenases. These don't exist and that literature needs to not be perpetuated.

Line 300 & 436: FHL not defined.

Line 340: I think H₂ likely does play a role in their metabolism, but this reads as a statement of certainty rather than one derived from gene homologies. Likewise, are FeFe hydrogenases more prominent? What is the reasoning for this.

Line 341: What is meant by FeFe hydrogenases being more active? Having higher rates of catalysis? This is generally true for H₂ production but not for H₂ oxidation. It is a major stretch to make such claims based on gene homology.

Line 350 and elsewhere: How do the authors know that these are membrane bound? Phylogenies only go so far, especially when considering weird divergent bugs like the authors are characterizing. Look for signal peptides, etc. that would give you more confidence in what you are suggesting.

Line 384: I think it is unlikely that bifurcating is energy conserving, *sensu strictu* since it does not form ATP and does not interconvert energy types like oxidative and substrate level phosphorylation do. I would delete this to avoid annoying a bioenergetics community member.

Line 413: Many early evolving phototrophs were photoheterotrophic (e.g., *Heliobacterium*).

Line 431: What is meant by an Ech having a respiratory function?

Line 436: How is this known based on genomic data?

Line 438: How is this known based on genomic data?

Line 442: Do they have antiporter domains?

Line 444: Ech do not produce energy. Rather, they function to interconvert differences in electrochemical potential into an ion gradient

Line 459: The only bifurcating NiFe hydrogenases associate with heterodisulfide reductase. There is no reason to speculate that these are bifurcating if they don't affiliate with these group of enzymes.

Line 464: One could say this about any enzyme. But the only way that metabolic models based on homology work is if they are founded on some biochemical framework. That this protein lacks the ligands for the active site metallocluster should be a strong indicator that it is not a NiFe hydrogenase. Again, see Schut et al., 2013 and 2016 for an idea of what these could be.

Line 474: Available evidence strongly indicates that this did happen and this continues to grow with new genome sequence data – see Schut 2016.

Line 490: Possess

Line 512: Do the authors mean acquired? If so, where might these enzyme homologs have come from?

Line 526: The authors do not know anything about the affinity of these proteins for O₂. They also don't know if they are specifically adapted to low O₂ levels.

Line 538-544: The authors do not address where the defining characteristics of Cyanobacteria come from? One could easily argue that the "siblings" of Cyanobacteria just acquired their anaerobic suite of enzymes via LGT, just like Cyanos must have done for photosynthetic reaction

centers, chlorophyll biosynthesis, CO₂ fixation, etc. Only when these potential scenarios are compared and contrasted can one be weighted more heavily than the other.

Fig. 1C. RA1A not in tree, yet it is used as the representative within the paper.

Fig. 2: indicate reversibility for reversible reactions. Putative FHL reaction is almost certainly incorrect. The putative group 3b hydrogenase need not involve S compounds.

Fig. 3A: group 3c NiFe hydrogenase should be a bifurcating type, correct? H₂ is not low enough potential to reduce Fd alone. Yet, these enzymes don't have the HdrBC subunits. It is also interesting that the FHL is now not assuming H₂ as electron donor, as opposed to Fig. 2. Garcia Costas 2017 go over the EftAB and suggest a suite of residues in the flavoprotein that need to be in place for these complexes to bifurcate.

Fig. 4A: Not Sulfhydrogenase.

Response to reviewers' comments on "Hydrogen-based metabolism – an ancestral trait in lineages sibling to the Cyanobacteria"

Reviewer #1 (Remarks to the Author):

Caveat: I am a phylogenomicist with first-hand experience in bacterial metagenomics, cyanobacterial phylogenetics and microalgal respiratory and photosynthetic metabolisms. In contrast, I am not an expert in all the peculiarities of the bacterial metabolisms. Therefore, the extensive metabolic discussions found in this manuscript were a tough read for me, and even if the authors appear to be very knowledgeable, I am afraid that this article might be difficult to follow for the broad readership of Nature Communications. Maybe the authors could consider expanding their introduction and adding a more streamlined overview of their results to help the non-experts reader to figure out what to recall from their study (see below).

*Denis Baurain
University of Liège*

We thank Dr. Baurain for these suggestions. We are aware that the results section can be challenging to digest for a general reader and have modified the manuscript to some extent. However, we are concerned about removing details that are critical to explain the foundation for our inferences. We have expanded the introduction section, as suggested.

General comments:

In this manuscript, Carnevali and co-workers mined new and existing metagenome and single-cell genomes from bacterial phyla supposedly related to Cyanobacteria in order to infer the ancestral metabolism of the whole group. They first used two small phylogenomic supermatrices to confirm that the phyla under study are indeed related to Cyanobacteria and how they relate to each other, and then searched for genes involved in energetic metabolism, building single-gene (family) trees to support their sub-classification of the various genes recovered.

Generally speaking, the work is thorough and appears to be methodologically sound. The figures were prepared with care and are graphically appealing. Therefore, I do not think that any "experiment" should be redone.

Thank you for this positive evaluation.

My main concern is the legibility of the whole manuscript, which is quite complicated to follow for someone not highly acquainted with non-conventional energetic metabolism. In particular, the three first introductory paragraphs are more confusing than anything, e.g., mixing ETCs for respiration and photosynthesis, assimilating respiration to O₂ use and fermentation to a fallback for anaerobic conditions. Here are a few background questions that might help the authors to better introduce the metabolic concepts discussed in their study. What is energy conservation and what is high-energy metabolism? What is the difference between respiration and fermentation? What is anaerobic respiration? Are there anaerobic ETCs? Can fermentation and anaerobic respiration cohabit in a single organism? What are alternative respiratory complexes? What are electron-bifurcating complexes and how do they achieve energy conservation? What are the various electron donors and acceptors that can be found in Bacteria? What is H₂ metabolism and how is it articulated with fermentation and anaerobic respiration?

We appreciate these concerns and in an effort to clarify the context and key concepts underlying the story and to make it more appealing to the broader audience of Nature Communications, we rewrote the introduction following the suggestions.

Specific comments:

Abstract

** Two sentences are a too vague to be really useful: 60-61: genomic features suggest the capacity for metabolic fine-tuning under strictly anoxic conditions; 69-71: Oxyphotobacteria may have acquired the ability to use O₂ as a terminal electron acceptor under certain environmental conditions.*

** Please try to be more specific, especially for the last sentence.*

These statements have been removed.

Introduction

** Given the age of the GOE and the very wide distribution of respiratory systems, I am not convinced that any inference from modern organisms can really “constrain the metabolic platform into which complex ETCs evolved” [lines 104-107]. For example, Melainabacteria were first described from human gut microbiome, which is certainly not the original environment where they have evolved from their common ancestor with the other lineages of interest. Thus, I would like the authors to be more cautious when explaining how their observations of modern organisms can help determining the lifestyles of their common ancestors [lines 119-122].*

We do not agree that it is impossible to make inferences about metabolic evolution from information about modern day organisms and in most instances this all we have to work with. However, we agree with the need for caution and have accordingly adjusted our language in parts of the manuscript (**lines 1139-1141**). Melainabacteria were actually first described from both an aquifer environment and the human gut (in the same paper), and are now found in many different environments. In the current study we have made use of the expanded genome set to search for features that are common to these organisms, and also shared with related bacteria.

Results

** All this study rests upon the accuracy of the reconstruction of the metagenomes under study. Apparently, 4 Saganbacteria genomes are circularized [lines 133-135], which suggests that they are completely assembled and thus more susceptible to give a faithful picture of their gene content. What about the other genomes? For example, the co-occurrence of Ech hydrogenase and Rnf complex is deemed to be rare yet observed in one Riflemargulisbacteria (RA1A) [lines 220-222]. Similarly, the lone coxB in the same Riflemargulisbacteria is puzzling [lines 278-280]. A sentence giving the limits of the study in terms of binning accuracy would be useful here.*

The Rnf complexes and Ech hydrogenases are found not in a single Margulisbacteria RA1A genome, but also in three other related genomes that are estimated to be 94-96% complete (**Supplementary Tables 1 and 2**). To make this more clear, we indicated that RA1A is the representative of an ANI cluster at the beginning of this section (**lines 384-385**). The availability of multiple near complete genomes within this lineage provides strong support for metabolic inferences.

The mystery of the single coxB (subunit II of cytochrome oxidase) gene in Riflemargulisbacteria genomes is a persistent one! We have seen it in other bins.

** From the Results only, it is not clear if the phylogenomic analyses are new or not [line 153-156]. Methods are detailed but references are given in the Results. This should be clarified. Moreover, are the two datasets independent? Are we sure that the 56 universal single copy proteins do not include any of the 16 ribosomal proteins? Otherwise, introduce the second as a subset of the first. Regarding phylogenetic resolution, some nodes have no symbol in Figure 1a-1b (and many other figures); do they correspond to maximal support? If so, this should be stated in the figure legends.*

We thank the reviewer for this comment. The species trees were indeed generated in this study. We clarified this by removing the references.

Furthermore, the reviewer is right, the 16 ribosomal proteins are subset of the 56 universal proteins. In the revised version of our manuscript we now added this information to the text (**line: 342**): “Importantly, the affiliation of these lineages with the Cyanobacteria was consistent for phylogenetic trees based on 56 universal single copy proteins and 16 ribosomal proteins which are a subset 56 universal single copy proteins (**Fig. 1a, Supplementary Fig. 1, 2**)”.

Regarding the phylogenetic resolution, nodes without support do indeed correspond to maximal support. We now state this in the legend of **Figure 1**: “Support values are not shown at nodes with maximal support (posterior probabilities >0.97).”

** The analysis of shared protein families is laudable but suffers from some caveats that should be explained. OrthoFinder always creates hundreds of thousands of gene “families”, ranging from very large groups composed of dozens of proteins per genome to (numerous) singletons containing only one single protein from one single genome. From Figure 1c, it is clear that “family” is used in that technical meaning, since Oxyphotobacteria alone (34 genomes) have 120,000 families! In my opinion, singletons should be removed from the pangenome statistics (or at least counted separately). By the way, how (and which) were the representative cyanobacterial genomes selected [lines 698-699]? Some publicly available cyanobacterial genomes are quite contaminated by foreign DNA (and proteins) and this might have affected the conclusions by adding a background of non-cyanobacterial families [Cornet et al. 2018, PLOS ONE; <https://www.biorxiv.org/content/early/2018/04/15/301788>].*

We appreciate this comment. As suggested by this reviewer, we now distinguish between singletons (protein families which are not shared between different class/phylum-level lineages) and shared protein families in the revised version of **Figure 1**. We were not aware of contamination of cyanobacterial genomes in public databases. We now screened all the genomes which were part of the shared protein family comparison for contamination using CheckM (Parks et al., 2015), revealing that only one of the selected genomes was affected by contamination (*Acaryochloris marina*, IMG 2648501120). This genome was excluded from the analysis and the figure updated accordingly. The re-analysis does not change our conclusions (all changes in the shared protein families lineage comparisons were below 1%). Both, taxon selection and contamination screening are now mentioned in the main text (**lines: 1580-1582**): “Cyanobacterial genomes were selected based on IMG taxonomy strings; for each cyanobacterial genus one genome with a predicted contamination of below 5 % as based on CheckM (Parks et al., 2015) was included in the dataset (Supplementary Table 5a).” Furthermore, we now provide a new **Supplementary Table 5a** showing the cyanobacterial strains selected for shared protein family comparisons and their contamination estimates based on CheckM.

Regarding the way shared families are computed, I don't understand the details [lines 701-704]. What is 100% in Figure 1c? When does a family count as "shared"? As soon as one protein from one of the genomes composing the pangenome is found in the same OrthoFinder group as another protein from another pangenome? Finally, the incongruence between the phylogenetic relationships deduced from the supermatrices and the gene content similarities computed between pangenomes [lines 166-171] argue for being very cautious when inferring the gene content and the metabolism of the common ancestor (see comment above).

A percentage of 100 in **Fig. 1c** would mean that all protein families (excluding singletons, see below) are shared between the compared pangenomes and may or may not be shared with other class/phylum-level lineages in the analysis. A protein family was counted as shared as soon as a single protein in the lineage pangenome is in a protein family together with proteins from other lineage pangenomes. To make this clearer we extended the methods section to further explain how the average percentage of shared protein families was calculated (**lines: 1580-1590**): "For each class/phylum-level lineage all proteomes of members of the respective lineage were combined to pan-proteomes. Protein families were identified with OrthoFinder (Emms and Kelly, 2015). Protein families which contained only proteins from a single lineage were treated as singletons and excluded from the following analysis. For all possible lineage pairs the percentage of shared protein families was calculated as the total number of protein families the respective lineage shared with the other divided by the total number of protein families found in this lineage. As lineage pan-proteomes differed in size, the average of the bidirectional percentage of shared protein families was taken for each pair."

We were less surprised about incongruence of species tree and shared protein content. As mentioned in the text, genomes of Marinamargulisbacteria and some Cyanobacteria might have been shaped by similar aerobic lifestyles.

** Some protein complexes are discussed using different names. This makes life harder for the casual reader. Please try to be consistent throughout the text and the figures (e.g., NiFe hydrogenases from group 4f = putative = Ehr?; group 4e = Ech?).*

Based on revisions suggested by Reviewer # 2 and other members of the scientific community (in response to the bioRxiv publication), we now refer to the different hydrogenases by predicted function in this new version of the manuscript. For instance, group 4e NiFe hydrogenases (following Greening et al. 2016) are now referred to as Ech hydrogenases (energy-converting hydrogenases). We also separated group 3 from group 4 NiFe hydrogenases in order to add new reference sequences that would allow us to improve our inferences about function. Please see new **Figure 4a** (groups 1, 2 and 3) **and b** (group 4 and related complexes). We hope this makes it easier for the reader.

** The discussion about the architecture of V-type ATPase and Rnf operons [lines 222-229] suggests that HGT may have played a role in the emergence of ETCs. If so, should it be discussed here and does it affect inferences about ancestral metabolism?*

The discussion of the architecture of V-type ATPase and Rnf operons is minimal. We chose not to pursue these aspects of the story, and instead focus on the membrane-bound hydrogenases and hydrogenase like complexes as key examples of ancestral forms of respiratory complexes.

Discussion

** The first paragraph [lines 408-414] suffers from the same interpretational issues as those pointed out in the introduction. It is too affirmative in its inference of the ancestral metabolic*

background in which have emerged photosynthesis and aerobic respiration, considering the age of the GOE and the subsequent niche adaptations observed by the authors in the sister lineages of Oxyphotobacteria. I am not saying that the authors are wrong, but I would like them to better differentiate between results in contemporaneous organisms and inferences about the past.

We acknowledge this concern and have modified the discussion to emphasize the long time period that separates the evolutionary events and today (**lines 894-1138**).

** Maybe more provocatively, in terms of significance, what could have been the common ancestor of all these phyla other than what is hypothesized here, i.e., an anaerobe with a fermentation-based metabolism, before the advent of photosynthesis and aerobic respiration [lines 419-423]? The alternative is not clear to me... but this is likely due to my own ignorance on the bacterial metabolisms.*

We are unable to suggest an alternative for the bacterial ancestor, prior to the availability of O₂.

** The last sentence of the discussion [553-555] is not informative. It might be better to reformulate it to make it less obvious.*

We agree and have deleted this sentence.

Methods

** Is it correct that Margulisbacteria RA1A is one bin out of 361 bins assembled from 14 samples [lines 607-618]? If so, there is ample room for binning errors, as suggested above. Please, add a sentence about this when introducing such metagenomes. If not, clarify the origin of this metagenome.*

It is correct that the bin is one of 361 bins from this 14-sample dataset, however each sample was individually assembled and RA1A was obtained from sample AAC11 (~ 27 bins). To find all scaffolds that belong to the specific genome and to remove all potential contamination we used paired-end read information to connect scaffolds. In addition, we also made sure that the number of links on the two sides of each scaffold is equal, and that the coverage of all scaffolds that could be linked is proportional to the number of links (e.g. a scaffold with n links on both sides is expected to appear in n copies across the genome and its coverage is expected to be roughly n time the coverage of most scaffolds). This procedure was done manually and resulted with nearly all scaffolds having the same number of links on both sides and linked to other scaffolds in the genome. We also used CheckM to evaluate the genome's completeness and contamination however our manual curation approach is very strict and allows us to confirm that the bin does not contain foreign scaffolds. These details have now been added to the Methods section.

** Still about this genome, I fail to see how the mini-assembly procedure has managed to reduce the number of scaffolds from 147 to 111 without adding longer reads [lines 627-638]. Is it because the various assembly thresholds are somewhat lower during that procedure than during the original assembly with idba_ud?*

In some instances, we identified one-to-one connections between scaffolds which allowed us to merge them into a single scaffold. All assemblers rely on certain heuristics and these heuristics sometimes fail to identify cases in which scaffolds could be elongated and merged. In our

experience there is usually room for improvement after the assembly process finished and this was the case with the current genome.

The reduction in the number of scaffolds was mostly due to removal of short scaffolds whose taxonomic affiliation was not consistent with the majority. Please see comment below.

** How exactly was performed the phylogenetic affiliation of the contigs [lines 647-648]? What database could provide enough taxonomically reliable reference sequences for that task? How much contigs had to be removed (from the 111?) after that control?*

Taxonomy is assigned based on genomes previously identified in our internal database (<https://ggkbase.berkeley.edu/>). In this case, the closest relatives were RBX1. The only reduction in contig numbers was from 147 to 111. Taxonomic assignment to scaffolds works as follows. First, all proteins from all scaffolds were aligned against an in-house database of proteins from reference genomes. For each protein we identified the best hit and assigned it a species-level taxonomy based on this best hit. Next, we used a majority-voting approach to assign a taxonomy to a scaffold, starting at the domain level all the way to the species level (if possible). This process ends once we reach a level in which no "winner" taxon has more than 50% of the proteins assigned to it. The level above this one provides our taxonomy assignment for the scaffold. The methods have been revised to now include this information.

** Gene content comparison is not enough described (see comment above). Moreover, I could not find the list of the selected cyanobacterial genomes [lines 694-705].*

We now extended description of our approach (see previous response) and included a table with information on reference genomes used for gene content comparison (**Supplementary Table 5a, b**).

** The selection of Terrabacteria genomes for the phylogenomic tree is not that clear to me. I understand that genomes (?) were dereplicated based on RNA polymerase [lines 711-715]. But how large and from where was the non-reduced set of genomes? Besides, is it correct that representative genomes were selected based on the fact that their RNA pol had been retained as the cluster representative by cd-hit? If so, this is not that smart given that (i) the only criterion to be the cd-hit representative is being the longest (first seen) sequence of the cluster and that (ii) fast-evolving organisms (not the best choice for phylogenomic trees) often have lengthened protein sequences. Please clarify here. More generally, this part of the Methods is difficult to follow in terms of datasets and figures (see also comment below about Supplemental Figures).*

We have clarified our approach in the revised manuscript. The non-reduced dataset was derived from publicly available bacterial genomes in IMG (~70,000). We performed cd-hit on RNA pol and then selected the most complete genome based on total number of observed phylogenetic markers as cluster representative. Only if the by cd-hit selected cluster representative was the most complete genome in the cluster it was selected as representative genome. We then further removed genomes from well sampled clades to make tree building with PhyloBayes on a concatenated alignment of 56 proteins computationally tractable. We now extended the methods section to better describe this approach (**lines: 1602-1606**): "A de-replicated set of reference genomes was obtained from publicly available bacterial genomes in IMG/M by COG0086 clustering at 65% sequence similarity and further de-replication of overrepresented clades. Cluster-representatives with the greatest number of different conserved marker genes were used to build the species tree of the Terrabacteria." (**Supplementary Table 5b**)

** The authors are very honest in their description of the various single-gene tree-building strategies [lines 736-762]. This is laudable. However, I fail to see why methods vary so widely between the different trees: different species sampling strategies, different aligners, different phylogenetic inference packages and evolutionary models. Could this be justified or is it just the result of many authors working each on their own?*

Different methods were deemed appropriate, based on the gene tree under consideration.

Wording and typos:

** line 108 (and others): “sibling” is better written as “sister” in a phylogenetic context*

We prefer to adopt the word “sibling” as it is gender neutral.

** line 119: “aerobically” is written twice*

** line 219: Fig. 2a should be Fig. 3a*

** line 247: nincotinamide => nicotinamide*

** line 287: Saganbacteria “are” heterotrophic anaerobes should be “include” considering the previous sentence about both aerobic and anaerobic representatives*

** line 300: FHL should be spelled out in full before line 446*

** line 446: “the” is written twice*

** line 481: the end of the sentence has an odd grammar*

** line 490: posses => possess*

** line 860: there is an extra “or” before NifJ*

** line 868: define “close proximity”*

** line 869: puple => purple*

** line 917: (in the table) Trasnport => Transport*

The typos listed above were fixed or sentences containing these words were rewritten.

Figures:

** Figure 3a: it might be useful to add the “e” and “f” to the corresponding group 4 NiFe hydrogenases for clarity. Moreover, there are two asterisks (*) that are not explained in the legend.*

We are now using the functional denomination only (please see comment above).

** Figure 3b: why not to show H⁺/Na⁺ translocation explicitly?*

We have modified the figure (new **Figure 3**), as this was not included by oversight.

** Supplemental Figure 1b is identical to Supplemental Figure 2. I am pretty sure that this is the result of a cut-and-paste error.*

This has been fixed.

** Supplemental Figure 6: the zoomed part would gain in legibility by annotating the various phyla observed in the tree.*

The figure was fixed to improve legibility (new **Supplementary Figure 5**).

Reviewer #2 (Remarks to the Author):

The paper by Metheus Carnevali et al explores the functional composition of MAGs or SCGs affiliated with groups that branch sister to Cyanobacteria. The authors characterize the putative metabolisms of MAGs/SCGs to provide insight into the metabolism of the ancestor of Cyanobacteria. Results are used to suggest that the ancestor is a fermentative bacterium that likely relies on group 4 NiFe hydrogenases or RNF complexes to drive ion translocation and generate ATP. The paper is of general interest but suffers from poor organization and overinterpretation in places. The following list of major and minor comments may help to improve the paper.

We are pleased that the reviewer finds the work of interest. We acknowledge that the material is complex and there are many interconnected stories. We have tried to clarify the findings in the revised version. Furthermore, the results were reorganized into four major headings, two of which encompass the different potential mechanisms for generation of a membrane potential in Margulisbacteria, Saganbacteria and Melainabacteria.

The title is potentially misleading. I don't think it is common to call fermentative microorganisms hydrogen based. They are carbon based and generate H₂.

The original title is : "Hydrogen-based metabolism – an ancestral trait in lineages sibling to the Cyanobacteria". This does not state that all fermentative organisms are hydrogen based. Rather, it conveys the key idea of the probably central role of hydrogen in some fermentative organisms.

Roughly a quarter of microbial genomes encode for one or more hydrogenases and 75% of the genomes of anaerobes encode bifurcating enzymes. I don't see these characteristics as being particularly unique to these organisms, siblings of Cyanobacteria or not.

We acknowledge this, but note that the key findings relate primarily to the large number and variety of hydrogenases.

The paper generally suffers from difficulty in its organization and the use of acronyms or loosely defined definitions. For example, is Oxyphotobacteria really a replacement for Cyanobacteria and has this been widely accepted in the field. I see this referenced out of Fischers group where it is claimed that this group postdates the rise of O₂ – not sure how this could be making me ever more skeptical of this new name. Moreover, how does this term capture anoxygenic photosynthesis by Cyanobacteria, or even fermentative metabolism within Cyanos that the authors reference in their paper.

We agree with this comment and ourselves adopted the term Oxyphotobacteria with some reluctance. Given this reviewer comment, the name "Oxyphotobacteria" has been removed from the manuscript.

The authors discussion of nitrogenase could be improved, and if done correctly, extremely revealing. NifH is a poor predictor of the phylogeny of Mo-nitrogenases. This needs to be done with a concatenation of NifHDK. Two monophyletic lineages will emerge in that phylogenetic reconstruction, those NifHDK hosted by aerobes/facultative anaerobes and those from strict

anaerobes. It would be interesting to see where the NifHDK from this putative anaerobe and potential ancestral state of Cyanobacteria branch.

We agree with the reviewer that a tree for NifHDK would have a better phylogenetic resolution compared to NifH. Thus, we replaced the single protein NifH tree with a new concatenated NifHDK tree (new **Figure 6 and Supplementary Data 3**). We used HMMs for each of the three proteins and screened our new genomes and all genomes available in IMG with these models. Genomes which had significant hits for all three proteins were then used as a reference data set for tree construction based on a concatenated alignment of NifHDK. We added a new paragraph (**lines: 1658-1670**) “NifHDK phylogenetic tree” to the methods section.

We thank the reviewer for this suggestion, it allowed us to make interesting inferences regarding the nitrogenase in Riflemargulisbacteria (**lines 530-540 and 1159-1161**).

In line 81, the authors present a paradox – oxygenic phototrophs (Cyanobacteria) require O₂ to drive the biosynthesis of some of the components of their cellular machinery. Yet, this is written as though the only opportunities to take advantage of O₂ were made available after they evolved. So, what came first – the ability to drive O₂ formation or the ability to synthesize components that require O₂? Perhaps all of this occurred stepwise in some logical way from a fermentative ancestor? Lots would need to take place including acquisition of CO₂ fixation and chlorophyll biosynthesis componentry.

We addressed this paradox by providing a potential sequence of events in the introduction.

Minor comments

Line 78: There would have been O₂ prior to the GOE but it would not have accumulated in the atmosphere. It would have likely been available for use by microbes, however, unless the kinetics of abiotic reactions were just that fast.

We agree with the reviewer and have clarified the introduction accordingly (**line 85**).

Line 79: No respiratory processes prior to O₂ then? What about Fe reduction or the Mtr complex in methanogens?

The primary oxidant required to produce ferric iron for use in anaerobic respiration is O₂. Thus, the great oxidation event is marked by massive Fe₂⁺ oxidation that led to formation of the banded iron formations. The same is true for the production of oxidized nitrogen and sulfur species (e.g., nitrate, sulfate). Thus, it seems to us that a modern type of anaerobic respiratory chain probably assembled after the availability of oxygen and oxidized compounds formed by O₂. The Mtr complex is of course of interest from the perspective of oxidative phosphorylation, as are the complexes described in the current study.

Line 98: Energy is not conserved during the reduction of O₂, but rather from the conversion of reducing equivalents to an electrochemical potential during ion translocation.

We clarified the processes involved in energy conservation in the introduction (**lines 77-81**) and throughout the manuscript.

Line 181: Sodium

Fixed.

Lines 182: From where and how recently? This is actually critical to your paper – perhaps the ancestor was like a cyanobacterium and diverged to form these organisms comparatively recently by acquisition of a number of anaerobic processes.

This question relates to the source of the transferred genes. We cannot offer a confident answer to this question.

Line 188: All life uses Fd. There are just different forms with different potentials that allow Fd use in aerobes.

Correct. We pointed this out because it was interesting to see the redundancy of enzymes with the same function using ferredoxin or NADH as an electron acceptor in the same genome.

Line 202: The discovery of FeFe hydrogenase in lineages sister to Cyanobacteria and the suggestion of ancestral character is an overlooked point here. It has long been problematic to the hydrogenase community why Cyanobacteria only encode NiFe hydrogenase while the rest of bacteria encode FeFe and NiFe hydrogenases. The other group of oxygenic phototrophs, algae, use FeFe hydrogenases but not NiFe hydrogenases. If the authors are correct in the overall interpretation of this paper, it would suggest that FeFe hydrogenases were lost and NiFe were retained. This may have to do with the FeFe being bifurcating type (see caveat below) and these are never found in the genomes of aerobes (they won't work).

This is very interesting, thank you for pointing this out. We now mention this in the manuscript (**lines 632-633**).

Line 214: Converting.

Fixed.

Line 216: These can be in numerous cells. The Greening paper artificially breaks these up by taxonomy (rather than function) in their phylogenetic assignments.

We have clarified this (**line 495**).

Line 221: Just a comment – the presence of Ech and Rnf, plus numerous bifurcating complexes, really does suggest these organisms are living on the thermodynamic edge. Wonder if this was the trigger to evolve something more splendid metabolically? i.e., oxygenic photosynthesis.

We agree that this is an interesting idea. We concur that the innovation of numerous different mechanisms to harvest energy likely positioned the ancestral organism for subsequent innovations and have clarified this in the current manuscript. However, we chose not to add a comment related to the thermodynamic edge because this interesting idea did not arise from our analyses and we should not lay claim to it.

Line 232: These are not NiFe hydrogenases if they lack the distal and viscinal Cys pairs that ligate the NiFe active site cluster. See Schut et al., 2013 and Schut et al., 2016 for a quick primer on what they could be.

Thank you for highlighting the importance of the listed references. The difference between group 4f NiFe hydrogenases (Greening et al., 2016) and hydrogenase-related complexes, such as the energy-converting hydrogenase related (Ehr) complexes and the membrane-bound hydrogenase related complex (Mbx), was clarified in the results section. We have now added new analyses and reinterpreted the findings. In order to shed light on this issue, we separated group 3 NiFe hydrogenases (new **Figure 4a**) from group 4, and the new tree for group 4 NiFe hydrogenases (new **Figure 4b**) also includes other membrane-bound complexes that are evolutionarily related (e.g, NADH:Ubiquinone oxidoreductase subunit D (NuoD), hydrogenase-related complexes (Ehr and Mbx), and additional Mbh-type group 4 hydrogenases).

Line 243: A reference is needed for this statement

Reference was added.

Line 245: The authors need to do more than to place the catalytic subunit of FeFe hydrogenase (HydA) functionally than just phylogenetics ala Greening 2016. If it is bifurcating, it will have HydBC (possibly HydD) in the flanking gene environment (see Poudel et al., 2016) for a classification scheme that is accurate.

We now note the identification of HydBC as well as HydA and follow the scheme discussed by Poudel et al. 2016 to determine the potential role of the FeFe hydrogenases. Additionally, we included reference sequences (from Therien et al. 2017, below) in our phylogenetic tree (new **Supplementary Figure 4 and Data 5**) representing the variety of modular structures described by their work (**also Supplementary Text and Table 4**).

Line 250: Thieren et al., 2017 have suggested FeFe hydrogenase variants that are regulated by N availability and that are catalytically biased towards H2 oxidation.

Thank you for pointing out this study. We cannot determine the catalytic bias based on the available information, however the genomic location in proximity to the nitrogenase may indicate regulation by N availability. This is now noted in the manuscript (**line 628**).

Line 272 and elsewhere: Phylogenetic analyses do not confirm that this is a type A heme-copper oxygen reductase adapted to atmospheric O2 levels. This type of overinterpretation through phylogenetics and gene/inferred protein homology detracts from the paper and is unnecessary.

This statement was largely based on the literature, given the phylogenetic placement as a type A oxygen reductase. We have removed the statement.

Line 281: Are they common in marine Cyanobacteria? I believe so.

We wrote ... "we did not identify any hydrogenases in the Marinamargulisbacteria, probably due to their different habitat". We acknowledge that some Cyanobacteria have hydrogenases, but suggest that habitat influences whether or not they are present. For example, about 89% of the Cyanobacteria from the open ocean do not have hydrogenases (Puggioni et al. 2016). We have added a sentence to place our inference in the context of this finding (**lines 823-824**).

Line 296: These are not sulfhydrogenases. These don't exist and that literature needs to not be perpetuated.

The term is indeed widely used in the literature for Group 3b hydrogenases. In response to this comment, we replaced the term 'Sulphydrogenase' by NADP-coupled NiFe hydrogenases, noting that some can use H₂S as an electron donor.

Line 300 & 436: FHL not defined.

After confirming that the majority of the putative group 4f hydrogenases (Ehf) in these organisms are actually energy-conserving hydrogenase related complexes (Ehr), we reviewed the potential role of the hypothetical formate hydrogenlyase (FHL) complex and determined that these complexes may have a slightly different function. Therefore, we eliminated the use of this acronym and introduced the idea of a formate dehydrogenase-Ehr complex (FDH-Ehr).

Line 340: I think H₂ likely does play a role in their metabolism, but this reads as a statement of certainty rather than one derived from gene homologies. Likewise, are FeFe hydrogenases more prominent? What is the reasoning for this.

We have eliminated these phrases. Regarding 'prominence', we stated this because FeFe hydrogenases are more numerous in genomes of Melainobacteria than in Margulisbacteria and Saganbacteria.

Line 341: What is meant by FeFe hydrogenases being more active? Having higher rates of catalysis? This is generally true for H₂ production but not for H₂ oxidation. It is a major stretch to make such claims based on gene homology.

We have removed statements that suggest any knowledge of potential levels of activity (these were based on literature information).

Line 350 and elsewhere: How do the authors know that these are membrane bound? Phylogenies only go so far, especially when considering weird divergent bugs like the authors are characterizing. Look for signal peptides, etc. that would give you more confidence in what you are suggesting.

We had observed transmembrane domains domains predicted via Interpro, which allowed us to make such inferences. We now state this information in the methods (**line 1606**).

Line 384: I think it is unlikely that bifurcating is energy conserving, sensu strictu since it does not form ATP and does not interconvert energy types like oxidative and substrate level phosphorylation do. I would delete this to avoid annoying a bioenergetics community member.

Thank you for this suggestion. This has been done.

Line 413: Many early evolving phototrophs were photoheterotrophic (e.g., Heliobacterium).

We agree that evolution of photosynthesis may have been preceded by other forms of use of light energy (e.g., photoheterotrophy). However, we have no way to incorporate consideration of photoheterotrophy in our analyses of the evolution of groups sibling to the Cyanobacteria.

Line 431: What is meant by an Ech having a respiratory function?

We meant that the Ech complex might play a role in the creation of a membrane potential, given that cytoplasmic redox reactions associated with the membrane-bound complex are coupled to ion translocation across the membrane. However, we have removed mention of respiration.

Line 436: How is this known based on genomic data?

Line 438: How is this known based on genomic data?

These statements are based on detailed analysis of encoded genes, protein domains and reference to experimentally-characterized complexes. This has now been clarified in the methods (see comment above).

Line 442: Do they have antiporter domains?

We did not identify proper antiporter-like (Mrp) domains. However, prior studies have demonstrated ion translocation associated with Rnf complexes. For example, the literature indicates that the Rnf complex in *Acetobacterium woodii* translocates Na⁺ ions during caffeate respiration (Biegel et al., 2010). The Rnf complex in *Methanosarcina acetivorans* (Schlegel et al., 2012) and *Clostridium ljungdahlii* (Tremblay et al., 2012) also couple cytoplasmic redox reactions with ion-translocation. This work is now cited in the discussion section (**line 1179**).

Line 444: Ech do not produce energy. Rather, they function to interconvert differences in electrochemical potential into an ion gradient.

We agree. This is diagrammed in **Figure 3** and has been clarified in the text.

Line 459: The only bifurcating NiFe hydrogenases associate with heterodisulfide reductase. There is no reason to speculate that these are bifurcating if they don't affiliate with these group of enzymes.

We removed this line of interpretation.

Line 464: One could say this about any enzyme. But the only way that metabolic models based on homology work is if they are founded on some biochemical framework. That this protein lacks the ligands for the active site metallocluster should be a strong indicator that it is not a NiFe hydrogenase. Again, see Schut et al., 2013 and 2016 for an idea of what these could be.

We thank the reviewer for this comment, which led us to further analysis. The identity of these protein complexes was revised. Please see other comments above about Ehr complexes.

Line 474: Available evidence strongly indicates that this did happen and this continues to grow with new genome sequence data – see Schut 2016.

Thank you for pointing out this study. We have added this citation.

Line 490: Possess

Fixed.

Line 512: Do the authors mean acquired? If so, where might these enzyme homologs have come from?

We concur and changed “evolved” to acquired. We have no way to constrain how the machinery for O₂ utilization was acquired.

Line 526: The authors do not know anything about the affinity of these proteins for O₂. They also don't know if they are specifically adapted to low O₂ levels.

These statements were based on the literature; this has now been clarified.

Line 538-544: The authors do not address where the defining characteristics of Cyanobacteria come from? One could easily argue that the “siblings” of Cyanobacteria just acquired their anaerobic suite of enzymes via LGT, just like Cyanos must have done for photosynthetic reaction centers, chlorophyll biosynthesis, CO₂ fixation, etc. Only when these potential scenarios are compared and contrasted can one be weighted more heavily than the other.

We feel that a strength of the current study is the finding that not just Melainabacteria have suites of anaerobic enzymes, but rather many groups sibling to both Cyanobacteria and Melainabacteria have these enzymes. Thus, we infer that it is more likely that the ancestral state was anaerobic and fermentation-based. We have revised the discussion to make this case.

Fig. 1C. RA1A not in tree, yet it is used as the representative within the paper.

We thank the reviewer for pointing this out. RA1A is part of a 97% ANI cluster for which we only showed the cluster representative GW1B in the tree. In the revised version of **Figure 1** we now show all the ANI cluster members in the tree.

Fig. 2: indicate reversibility for reversible reactions. Putative FHL reaction is almost certainly incorrect.

Reversibility is now indicated in the new **Figures 2 and 3**. We redefined the role of a putative formate dehydrogenase-Ehr complex and it is now described in the text.

The putative group 3b hydrogenase need not involve S compounds.

We have the terminology, as noted in the response above.

Fig. 3A: group 3c NiFe hydrogenase should be a bifurcating type, correct? H₂ is not low enough potential to reduce Fd alone. Yet, these enzymes don't have the HdrBC subunits.

We agree with this comment. The complex was shown based on the proteins identified. We were uncertain of the partner, given that we did not identify HdrBC subunits. We have modified the figure to indicate uncertainty as to the nature of other components that would be required for function (new **Figures 2 and 3**).

It is also interesting that the FHL is now not assuming H₂ as electron donor, as opposed to Fig. 2.

This was an oversight in the original manuscript. We now do not believe that the complex is a FHL and have revised the manuscript throughout.

Garcia Costas 2017_go over the EftAB and suggest a suite of residues in the flavoprotein that need to be in place for these complexes to bifurcate.

We have removed reference to bifurcation after reexamination of the domain composition. We now suggest that they have an AMP-binding domain rather than a second flavoprotein-binding domain.

Fig. 4A: Not Sulphydrogenase.

We have removed the terminology, as noted in the response above.

REVIEWERS' COMMENTS:

Reviewer #1 (Remarks to the Author):

Thank you for your thorough revision of this manuscript. It still is a tough read for me, but I guess this is unavoidable due to its topic. Therefore, I think it is now ready for publication. I only have very minor residual comments that you will be able to address swiftly.

* Curation of genomes derived from metagenomes

Thank you for explaining your method of taxonomic assignment. I think it is sound. My only concern is that suitable reference genomes might be lacking. In other words, do you have trustful closely related genomes for the new lineages that you are curating in your study? I guess that many of the cyanobacterial neighbors are only available as metagenomic assemblies. Do you include these in your in-house database (line 1438)? If so, are there trustful? Otherwise, how do you assess that your scaffolds are of the expected taxonomy? Please add a sentence to clarify the issue.

* Gene content comparison

I much appreciate how you clarified your whole procedure. However, I am still puzzled about what would be 100%. Is it the union of all the non-singleton families of the two lineages under comparison (hence a different number for each pair) or is it the union of all the non-singleton families of the whole set of lineages (a true unique 100%)? Or is it something else (due to your approach of taking the average of the bidirectional percentage)? Please, add a few words to define this properly.

* NifHDK phylogenetic tree

You seem to use two rounds of HMM searches. I can see the point if you think that your initial pHMMs are not broad enough, hence the first round on the IMG/M database to expand their taxonomic breadth. However, your subsequent sentences let the impression that these improved profiles are then used on the exact same database. Am I mistaken? Please check that your wording is accurate. It might make sense but this should be better explained then. Moreover, you seem to dereplicate HMM hits until the very last step. If this is done on individual HMM result sets, how do you decide which hits to concatenate to avoid creating chimeras? Please clarify.

* Typos:

- lines 61, 62 (and others): O2 is typeset with a zero instead of the letter O
- line 634: Saganbacteria have (not has) anaerobic...
- line 647: hydrogeanse => hydrogenase
- line 653: "Figure" is missing
- line 816: and a (not an) type A...
- line 915: encode or code for (but not encode for, I guess)
- line 1463: used to fill up (not full) gaps...

In the Supplementary Information, you still have two mentions to Oxyphotobacteria (lines 361 and 362) and to sulfhydrogenase (lines 122 and 127) that you should probably edit to be consistent with your main text.

Reviewer #2 (Remarks to the Author):

The authors have made a comprehensive attempt at revising their manuscript. I only have a few additional edits to consider that are aimed at further improving their final product.

Line 388: "balance reducing equivalent pools"

Line 536: should be nifENBV that are involved in synthesis

Line 625: "and its expression could be ..."

Line 742: replace "is" with "can be"

Line 786: replace "possible" with "likely in this complex"

Line 915 and elsewhere: "code for" or "encode" but not "encode for" since this is redundant

Line 1100-1107: these enzymes do not conserve energy but rather interconvert energy between ionic gradients and electrical potential. This paragraph could use some work to make sure this is made clear to the reader. For example, the MBH in *T. onnurineus* creates an H⁺/Na⁺ gradient by oxidizing Fd coupled to production of H₂.

Line 1180: delete "the" preceding Nfn

Line 1181: specify the classification scheme. if Vignais's scheme, then group 3d are not bifurcating but rather are bidirectional.

Line 1183: Why are they notable?

Response to referees

Reviewer #1 (Remarks to the Author):

Thank you for your thorough revision of this manuscript. It still is a tough read for me, but I guess this is unavoidable due to its topic. Therefore, I think it is now ready for publication. I only have very minor residual comments that you will be able to address swiftly.

** Curation of genomes derived from metagenomes*

Thank you for explaining your method of taxonomic assignment. I think it is sound. My only concern is that suitable reference genomes might be lacking. In other words, do you have trustful closely related genomes for the new lineages that you are curating in your study? I guess that many of the cyanobacterial neighbors are only available as metagenomic assemblies. Do you include these in your in-house database (line 1438)? If so, are there trustful? Otherwise, how do you assess that your scaffolds are of the expected taxonomy? Please add a sentence to clarify the issue.

We appreciate the concerns raised by reviewer #1. As the reviewer points out most of the genomes included are derived from metagenomes, but many of the resulting reference genomes are of very high-quality having passed numerous quality controls checks for genome completeness and contamination in our in-house database (ggDB). Our database includes isolates from public databases (NCBI), high-quality published MAGs (NCBI, IMG/M), and unpublished MAGs reconstructed in our laboratory. Be believe that this represents a comprehensive reference dataset of high quality and to be trusted for robust phylogenetic analyses.

Additionally, our expected taxonomy from the specific scaffolds (and genomes) come from different environments and in some cases, represent circular genomes. As an example, 4 Saganbacteria genomes recovered from subsurface sediments at Rifle (Colorado) are complete and circular, and their taxonomy matches with Saganbacteria MAGs recovered from the Crystal Geyser aquifer in Utah and the White Oak river estuary in North Carolina.

In response to this comment, we now added a couple of sentences to further clarify this issue (**Supplementary Methods lines 99-102**).

** Gene content comparison*

I much appreciate how you clarified your whole procedure. However, I am still puzzled about what would be 100%. Is it the union of all the non-singleton families of the two lineages under comparison (hence a different number for each pair) or is it the union of all the non-singleton families of the whole set of lineages (a true unique 100%)? Or is it something else (due to your approach of taking the average of the bidirectional percentage)? Please, add a few words to define this properly.

The reviewer's first suggestion is correct: 100% would represent the union of all the non-singleton families of the two lineages under comparison. To make this clearer to the reader we now added this information to the legend of figure 1.

** NifHDK phylogenetic tree*

You seem to use two rounds of HMM searches. I can see the point if you think that your initial pHMMs are not broad enough, hence the first round on the IMG/M database to expand their taxonomic breadth. However, your subsequent sentences let the impression that these improved profiles are then used on the exact same database. Am I mistaken? Please check that your wording is accurate. It might make sense but this should be better explained then. Moreover, you seem to dereplicate HMM hits until the very last step. If this is done on individual HMM result sets, how do you decide which hits to concatenate to avoid creating chimeras? Please clarify.

The reviewer is correct, the improved models were used to screen the new genomes generated in this study and again all genomes in IMG/M, making this an iterative process. To make this clearer in the manuscript, we now changed our wording in the methods section: "The improved HMMs were then used to identify NifHDK in novel genomes and microbial genomes available in the IMG/M system using hmmsearch". Regarding the reviewer's second question, de-replication was performed based on clustering of NifK, which was the in average longest protein out of NifHDK. We now state this more concisely in the methods section: "Protein hits were extracted and de-replicated based on clustering of NifK at 90% sequence similarity with MCL. NifHDK of cluster medoids were then aligned with mafft, and alignments trimmed with trimal to remove positions with more than 90% gaps."

** Typos:*

- lines 61, 62 (and others): O2 is typeset with a zero instead of the letter O*
- line 634: Saganbacteria have (not has) anaerobic...*
- line 647: hydrogeanse => hydrogenase*
- line 653: "Figure" is missing*
- line 816: and a (not an) type A...*
- line 915: encode or code for (but not encode for, I guess)*
- line 1463: used to fill up (not full) gaps...*

The typos listed above were corrected.

In the Supplementary Information, you still have two mentions to Oxyphotobacteria (lines 361 and 362) and to sulfhydrogenase (lines 122 and 127) that you should probably edit to be consistent with your main text.

Oxyphotobacteria was replaced by Cyanobacteria and sulfhydrogenase was replaced by NADP-reducing NiFe hydrogenase.

Reviewer #2 (Remarks to the Author):

The authors have made a comprehensive attempt at revising their manuscript. I only have a few additional edits to consider that are aimed at further improving their final product.

Line 388: "balance reducing equivalent pools"

Line 536: should be nifENBV that are involved in synthesis

Line 625: "and its expression could be ..."

Line 742: replace "is" with "can be"

Line 786: replace "possible" with "likely in this complex"

Line 915 and elsewhere: "code for" or "encode" but not "encode for" since this is redundant

Line 1180: delete "the" preceding Nfn

Words and sentences listed above were corrected.

*Line 1100-1107: these enzymes do not conserve energy but rather interconvert energy between ionic gradients and electrical potential. This paragraph could use some work to make sure this is made clear to the reader. For example, the MBH in *T. onnurineus* creates an H^+/Na^+ gradient by oxidizing Fd coupled to production of H_2 .*

This paragraph (**lines 572-589**) was revised accordingly, to clarify the role of membrane-bound NiFe hydrogenases.

Line 1181: specify the classification scheme. if Vignais's scheme, then group 3d are not bifurcating but rather are bidirectional.

Following Vignais's scheme group 3d NiFe hydrogenases are indeed bidirectional, therefore they were removed from the list.

Line 1183: Why are they notable?

This sentence was revised to highlight that electron bifurcation is common in anaerobic organisms (**lines 617-619**).